# Nonstationary Reinforcement Learning with Linear Function Approximation

**Huozhi Zhou**  *hzhou35@illinois.edu*
*Department of Electrical and Computer Engineering*
*University of Illinois Urbana-Champaign*

**Jinglin Chen**  *jinglinc@illinois.edu*
*Department of Computer Science*
*University of Illinois Urbana-Champaign*

**Lav R. Varshney**  *varshney@illinois.edu*
*Department of Electrical and Computer Engineering*
*University of Illinois Urbana-Champaign*

**Ashish Jagmohan**  *ashishja@us.ibm.com*
*IBM Research*

**Reviewed on OpenReview:** *https: // openreview. net/ forum? id= nS8A9nOrqp*

## Abstract

We consider reinforcement learning (RL) in episodic Markov decision processes (MDPs) with linear function approximation under drifting environment. Specifically, both the reward and state transition functions can evolve over time but their total variations do not exceed a *variation budget*. We first develop `LSVI-UCB-Restart` algorithm, an optimistic modification of least-squares value iteration with periodic restart, and bound its dynamic regret when variation budgets are known. Then we propose a parameter-free algorithm `Ada-LSVI-UCB-Restart` that extends to unknown variation budgets. We also derive the first minimax dynamic regret lower bound for nonstationary linear MDPs and as a byproduct establish a minimax regret lower bound for linear MDPs unsolved by Jin et al. (2020). Finally, we provide numerical experiments to demonstrate the effectiveness of our proposed algorithms.

## 1 Introduction

Reinforcement learning (RL) is a core control problem in which an agent sequentially interacts with an unknown environment to maximize its cumulative reward (Sutton & Barto, 2018). RL finds enormous applications in real-time bidding in advertisement auctions (Cai et al., 2017), autonomous driving (Shalev-Shwartz et al., 2016), gaming-AI (Silver et al., 2018), and inventory control (Agrawal & Jia, 2019), among others. Due to the large dimension of sequential decision-making problems that are of growing interest, classical RL algorithms designed for finite state space such as `tabular Q-learning` (Watkins & Dayan, 1992) no longer yield satisfactory performance. Recent advances in RL rely on function approximators such as deep neural nets to overcome the curse of dimensionality, i.e., the value function is approximated by a function which is able to predict the value function for unseen state-action pairs given a few training samples. This function approximation technique has achieved remarkable success in various large-scale decision-making problems such as playing video games (Mnih et al., 2015), the game of Go (Silver et al., 2017), and robot control (Akkaya et al., 2019). Motivated by the empirical success of RL algorithms with function approximation, there is growing interest in developing RL algorithms with function approximation that are statistically efficient (Yang & Wang, 2019; Cai et al., 2020; Jin et al., 2020; Modi et al., 2020; Wang

et al., 2020; Wei et al., 2021; Neu & Olkhovskaya, 2021; Jiang et al., 2017; Wang et al., 2020; Jin et al., 2021; Du et al., 2021). The focus of this line of work is to develop statistically efficient algorithms with function approximation for RL in terms of either *regret* or *sample complexity*. Such efficiency is especially crucial in data-sparse applications such as medical trials (Zhao et al., 2009).

However, all of the aforementioned empirical and theoretical works on RL with function approximation assume the environment is stationary, which is insufficient to model problems with time-varying dynamics. For example, consider online advertising. The instantaneous reward is the payoff when viewers are redirected to an advertiser, and the state is defined as the the details of the advertisement and user contexts. If the target users' preferences are time-varying, time-invariant reward and transition function are unable to capture the dynamics. In general nonstationary random processes naturally occur in many settings and are able to characterize larger classes of problems of interest (Cover & Pombra, 1989). Can one design a theoretically sound algorithm for large-scale nonstationary MDPs? In general it is impossible to design algorithm to achieve sublinear regret for MDPs with non-oblivious adversarial reward and transition functions in the worst case (Yu et al., 2009). Then what is the maximum *nonstationarity* a learner can tolerate to adapt to the time-varying dynamics of an MDP with potentially infinite number of states? This paper addresses these two questions.

We consider the setting of episodic RL with nonstationary reward and transition functions. To measure the performance of an algorithm, we use the notion of *dynamic regret*, the performance difference between an algorithm and the set of policies optimal for individual episodes in hindsight. For nonstationary RL, dynamic regret is a stronger and more appropriate notion of performance measure than static regret, but is also more challenging for algorithm design and analysis. To incorporate function approximation, we focus on a subclass of MDPs in which the reward and transition dynamics are linear in a known feature map (Melo & Ribeiro, 2007), termed *linear MDP*. For any linear MDP, the value function of any policy is linear in the known feature map since the Bellman equation is linear in reward and transition dynamics (Jin et al., 2020). Since the optimal policy is greedy with respect to the optimal value function, linear function approximation suffices to learn the optimal policy. For nonstationary linear MDPs, we show that one can design a near-optimal statistically-efficient algorithm to achieve sublinear dynamic regret as long as the total variation of reward and transition dynamics is sublinear. Let $T$ be the total number of time steps, $B$ be the total variation of reward and transition function throughout the entire time horizion, $d$ be the ambient dimension of the features, and $H$ be the planning horizon.

The contribution of our work is summarized as follows.

- We prove a $\Omega(B^{1/3}d^{2/3}H^{1/3}T^{2/3})$ minimax regret lower bound for nonstationary linear MDP, which shows that it is impossible for any algorithm to achieve sublinear regret on any nonstationary linear MDP with total variation linear in $T$. As a byproduct, we also derive the minimax regret lower bound for stationary linear MDP on the order of $\Omega(d\sqrt{HT})$, which is unsolved in Jin et al. (2020).

- We develop the `LSVI-UCB-Restart` algorithm and analyze the dynamic regret bound for both cases that local variations are known or unknown, assuming the total variations are known. We define local variations (Eq. (2)) as the change in the environment between two consecutive epochs instead of the total changes over the entire time horizon. When local variations are known, `LSVI-UCB-Restart` achieves $\tilde{O}(B^{1/3}d^{4/3}H^{4/3}T^{2/3})$ dynamic regret, which matches the lower bound in $B$ and $T$, up to polylogarithmic factors. When local variations are unknown, `LSVI-UCB-Restart` achieves $\tilde{O}(B^{1/4}d^{5/4}H^{5/4}T^{3/4})$ dynamic regret.

- We propose a parameter-free algorithm called `Ada-LSVI-UCB-Restart`, an adaptive version of `LSVI-UCB-Restart`, and prove that it can achieve $\tilde{O}(B^{1/4}d^{5/4}H^{5/4}T^{3/4})$ dynamic regret without knowing the total variations.

- We conduct numerical experiments on synthetic nonstationary linear MDPs to demonstrate the effectiveness of our proposed algorithms.

## 1.1 Related Works

**Nonstationary bandits**  Bandit problems can be viewed as a special case of MDP problems with unit planning horizon. It is the simplest model that captures the exploration-exploitation tradeoff, a unique feature of sequential decision-making problems. There are several ways to define nonstationarity in the bandit literature. The first one is piecewise-stationary (Garivier & Moulines, 2011), which assumes the expected rewards of arms change in a piecewise manner, i.e., stay fixed for a time period and abruptly change at unknown time steps. The second one is to quantify the total variations of expected rewards of arms (Besbes et al., 2014). The general strategy to adapt to nonstationarity for bandit problems is the forgetting principle: run the algorithm designed for stationary bandits either on a sliding window or in small epochs. This seemingly simple strategy is successful in developing near-optimal algorithms for many variants of nonstationary bandits, such as cascading bandits (Wang et al., 2019), combinatorial semi-bandits (Zhou et al., 2020) and linear contextual bandits (Cheung et al., 2019; Zhao et al., 2020; Russac et al., 2019). Other nonstationary bandit models include the nonstationary rested bandit, where the reward of each arm changes only when that arm is pulled (Cortes et al., 2017), and online learning with expert advice (Mohri & Yang, 2017a;b), where the qualities of experts are time-varying. However, reinforcement learning is much more intricate than bandits. Note that naïvely adapting existing nonstationary bandit algorithms to nonstationary RL leads to regret bounds with exponential dependence on the planing horizon $H$.

**RL with function approximation**  Motivated by empirical success of deep RL, there is a recent line of work analyzing the theoretical performance of RL algorithms with function approximation (Yang & Wang, 2019; Cai et al., 2020; Jin et al., 2020; Modi et al., 2020; Ayoub et al., 2020; Wang et al., 2020; Zhou et al., 2021; Wei et al., 2021; Neu & Olkhovskaya, 2021; Huang et al., 2021; Modi et al., 2021; Jiang et al., 2017; Agarwal et al., 2020; Dong et al., 2020; Jin et al., 2021; Du et al., 2021; Foster et al., 2021a; Chen et al., 2022). Recent work also studies the instance-dependent sample complexity bound for RL with function approximation, which adapts to the complexity of the specific MDP instance (Foster et al., 2021b; Dong & Ma, 2022). All of these works assume that the learner is interacting with a stationary environment. In sharp contrast, this paper considers learning in a nonstationary environment. As we will show later, if we do not properly adapt to the nonstationarity, linear regret is incurred.

**Nonstationary RL**  The last relevant line of work is on dynamic regret analysis of nonstationary MDPs mostly without function approximation (Auer et al., 2010; Ortner et al., 2020; Cheung et al., 2019; Fei et al., 2020; Cheung et al., 2020). The work of Auer et al. (2010) considers the setting in which the MDP is piecewise-stationary and allowed to change in $l$ times for the reward and transition functions. They show that UCRL2 with restart achieves $\tilde{O}(l^{1/3}T^{2/3})$ dynamic regret, where $T$ is the time horizon. Later works (Ortner et al., 2020; Cheung et al., 2020; Fei et al., 2020) generalize the nonstationary setting to allow reward and transition functions vary for any number of time steps, as long as the total variation is bounded. Specifically, the work of (Ortner et al., 2020) proves that UCRL with restart achieves $\tilde{O}((B_r + B_p)^{1/3}T^{2/3})$ dynamic regret (when the variation in each epoch is known), where $B_r$ and $B_p$ denote the total variation of reward and transition functions over all time steps. Cheung et al. (2020) proposes an algorithm based on UCRL2 by combining sliding windows and a confidence widening technique. Their algorithm has slightly worse dynamic regret bound $\tilde{O}((B_r + B_p)^{1/4}T^{3/4})$ without knowing the local variations. Further, Fei et al. (2020) develops an algorithm which directly optimizes the policy and enjoys near-optimal regret in the low-variation regime. A different model of nonstationary MDP is proposed by Lykouris et al. (2021), which smoothly interpolates between stationary and adversarial environments, by assuming that most episodes are stationary except for a small number of adversarial episodes. Note that Lykouris et al. (2021) considers linear function approximation, but their nonstationarity assumption is different from ours. In this paper, we assume the variation budget for reward and transition function is bounded, which is similar to the settings in Ortner et al. (2020); Cheung et al. (2020); Mao et al. (2021). Concurrently to our work, Touati & Vincent (2020) propose an algorithm combining weighted least-squares value iteration and the optimistic principle, achieving the same $\tilde{O}(B^{1/4}d^{5/4}H^{5/4}T^{3/4})$ regret as we do with knowledge of the total variation $B$. They do not have a dynamic regret bound when the knowledge of local variations is available. Their proposed algorithm uses exponential weights to smoothly forget data that are far in the past. By contrast, our algorithm periodically restarts the LSVI-UCB algorithm from scratch to handle the non-stationarity and is much more computationally

efficient. Another concurrent work by Wei & Luo (2021) follows a substantially different approach to achieve the optimal $T^{2/3}$ regret. The key idea of their algorithm is to run multiple base algorithms for stationary instances with different duration simultaneously, under a carefully designed random schedule. Compared with them, our algorithm has a slightly worse rate, but a much better computational complexity, since we only require to maintain one instance of the base algorithm. Both of these two concurrent works do not have empirical results, and we are also the first one to conduct numerical experiments on online exploration for non-stationary MDPs (Section 6). Other related and concurrent works investigate online exploration in different classes of non-stationary MDPs, including linear kernal MDP (Zhong et al., 2021), constrained tabular MDP (Ding & Lavaei, 2022), and stochastic shorted path problem (Chen & Luo, 2022).

The rest of the paper is organized as follows. Section 2 presents our problem definition. Section 3 establishes the minimax regret lower bound for nonstationary linear MDPs. Section 4 and Section 5 present our algorithms LSVI-UCB-Restart, Ada-LSVI-UCB-Restart and their dynamic regret bounds. Section 6 shows our experiment results. Section 7 concludes the paper and discusses some future directions. All detailed proofs can be found in Appendices.

**Notation** We use $\langle \cdot, \cdot \rangle$ to denote inner products in Euclidean space, $\|\boldsymbol{v}\|_2$ to denote the $L_2$ norm of vector $\boldsymbol{v}$, and $\|\boldsymbol{v}\|_\Lambda$ to denote the norm induced by a positive definite matrix $A$ for vector $\boldsymbol{v}$, i.e., $\|\boldsymbol{v}\|_\Lambda = \sqrt{\boldsymbol{v}^\top \Lambda \boldsymbol{v}}$. For an integer $N$, we denote the set of positive integers $\{1, 2, \ldots, N\}$ as $[N]$.

## 2 Preliminaries

We consider the setting of a nonstationary finite-horizon episodic Markov decision process (MDP), specified by a tuple $(\mathcal{S}, \mathcal{A}, H, K, \mathbb{P} = \{\mathbb{P}_h^k\}_{h \in [H], k \in [K]}, r = \{r_h^k\}_{h \in [H], k \in [K]})$, where the set $\mathcal{S}$ is the collection of states, $\mathcal{A}$ is the collection of actions, $H$ is the length of each episode, $K$ is the total number of episodes, and $\mathbb{P}$ and $r$ are the transition kernel and deterministic reward functions respectively. Moreover, $\mathbb{P}_h^k(\cdot|s, a)$ denotes the transition kernel over the next states if the action $a$ is taken for state $s$ at step $h$ in the $k$-th episode, and $r_h^k : \mathcal{S} \times \mathcal{A} \to [0, 1]$ is the deterministic reward function at step $h$ in the $k$-th episode. Note that we are considering a nonstationary setting, thus we assume the transition kernel $\mathbb{P}$ and reward function $r$ may change in different episodes. We will explicitly quantify the nonstationarity later.

The learning protocol proceeds as follows. In episode $k$, the initial state $s_1^k$ is chosen by an adversary. Then at each step $h \in [H]$, the agent observes the current state $s_h \in \mathcal{S}$, takes an action $a_h \in \mathcal{A}$, receives an instantaneous reward $r_h^k(s_h, a_h)$, then transitions to the next state $s_{h+1}$ according to the distribution $\mathbb{P}_h^k(\cdot|s_h, a_h)$. This process terminates at step $H$ and then the next episode begins. The agent interacts with the environment for $K$ episodes, which yields $T = KH$ time steps in total.

A policy $\pi$ is a collection of functions $\pi_h : \mathcal{S} \to \mathcal{A}, \forall h \in [H]$. We define the value function at step $h$ in the $k$-th episode for policy $\pi$ as the expected value of the cumulative rewards received under policy $\pi$ when starting from an arbitrary state $s$:

$$V_{h,k}^\pi(s) = \mathbb{E}_\pi \left[ \sum_{h'=h}^{H} r_{h'}^k(s_{h'}, a_{h'}) | s_h = s \right], \forall s \in \mathcal{S}, h \in [H], k \in [K].$$

Note that the value function also depends on the episode $k$ since the Markov decision process is nonstationary. Similarly, we can also define the action-value function (a.k.a. $Q$ function) for policy $\pi$ at step $h$ in the $k$-th episode, which gives the expected cumulative reward starting from an arbitrary state-action pair:

$$Q_{h,k}^\pi(s, a) = r_h^k(s, a) + \mathbb{E}_\pi \left[ \sum_{h'=h+1}^{H} r_{h'}^k(s_{h'}, a_{h'}) | s_h = s, a_h = a \right], \forall (s, a) \in \mathcal{S} \times \mathcal{A}, h \in [H], k \in [K].$$

We define the optimal value function and optimal action-value function for step $h$ in $k$-th episode as $V_{h,k}^*(s) = \sup_\pi V_{h,k}^\pi(s)$ and $Q_{h,k}^*(s, a) = \sup_\pi Q_{h,k}^\pi(s, a)$ respectively, which always exist (Puterman, 2014). For simplicity, we denote $\mathbb{E}_{s' \sim \mathbb{P}_h^k(\cdot|s,a)}[V_{h+1}(s')] = [\mathbb{P}_h^k V_{h+1}](s, a)$. Using this notation, we can write down the

Bellman equation for any policy $\pi$,

$$Q_{h,k}^{\pi}(s,a) = (r_h^k + \mathbb{P}_h^k V_{h+1,k}^{\pi})(s,a), V_{h,k}^{\pi}(s) = Q_{h,k}^{\pi}(s,\pi_h(s)), V_{H+1,k}^{\pi}(s) = 0, \forall (s,a,k) \in \mathcal{S} \times \mathcal{A} \times [K].$$

Similarly, the Bellman optimality equation is

$$Q_{h,k}^{*}(s,a) = (r_h^k + \mathbb{P}_h^k V_{h+1,k}^{*})(s,a), V_{h,k}^{*}(s) = \max_{a \in \mathcal{A}} Q_{h,k}^{*}(s,a), V_{H+1,k}^{*}(s) = 0, \forall (s,a,k) \in \mathcal{S} \times \mathcal{A} \times [K].$$

This implies that the optimal policy $\pi^*$ is greedy with respect to the optimal action-value function $Q^*(\cdot,\cdot)$. Thus in order to learn the optimal policy it suffices to estimate the optimal action-value function.

Recall that the agent is learning the optimal policy via interactions with the environment, despite the uncertainty of $r$ and $\mathbb{P}$. In the $k$-th episode, the adversary chooses the initial state $s_1^k$, then the agent decides its policy $\pi^k$ for this episode based on historical interactions. To measure the convergence to optimality, we consider an equivalent objective of minimizing the *dynamic regret* (Cheung et al., 2020),

$$\text{Dyn-Reg}(K) = \sum_{k=1}^{K} \left[ V_{1,k}^{*}(s_1^k) - V_{1,k}^{\pi^k}(s_1^k) \right]. \tag{1}$$

## 2.1 Linear Markov Decision Process

We consider a special class of MDPs called *linear Markov decision process* (Melo & Ribeiro, 2007; Bradtke & Barto, 1996; Jin et al., 2020), which assumes both transition function $\mathbb{P}$ and reward function $r$ are linear in a known feature map $\phi(\cdot,\cdot)$. The formal definition is as follows.

**Definition 1.** *(Linear MDP). The MDP $(\mathcal{S}, \mathcal{A}, H, K, \mathbb{P}, r)$ is a linear MDP with the feature map $\phi : \mathcal{S} \times \mathcal{A} \to \mathbb{R}^d$, if for any $(h,k) \in [H] \times [K]$, there exist $d$ unknown measures $\boldsymbol{\mu}_{h,k} = (\boldsymbol{\mu}_{h,k}^1, \ldots, \boldsymbol{\mu}_{h,k}^d)^\top$ on $\mathcal{S}$ and a vector $\boldsymbol{\theta}_{h,k} \in \mathbb{R}^d$ such that*

$$\mathbb{P}_h^k(s'|s,a) = \phi(s,a)^\top \boldsymbol{\mu}_{h,k}(s'), \quad r_h^k(s,a) = \phi(s,a)^\top \boldsymbol{\theta}_{h,k}.$$

*Without loss of generality, we assume $\|\phi(s,a)\|_2 \le 1$ for all $(s,a) \in \mathcal{S} \times \mathcal{A}$, and $\max\{\|\boldsymbol{\mu}_{h,k}(\mathcal{S})\|_2, \|\boldsymbol{\theta}_{h,k}\|_2\} \le \sqrt{d}$ for all $(h,k) \in [H] \times [K]$.*

Note that the transition function $\mathbb{P}$ and reward function $r$ are determined by the unknown measures $\{\boldsymbol{\mu}_{h,k}\}_{h \in [H], k \in [K]}$ and latent vectors $\{\boldsymbol{\theta}_{h,k}\}_{h \in [H], k \in [K]}$. The quantities $\boldsymbol{\mu}_{h,k}$ and $\boldsymbol{\theta}_{h,k}$ vary across time in general, which leads to change in transition function $\mathbb{P}$ and reward function $r$. Following Besbes et al. (2014); Cheung et al. (2019; 2020), we quantify the total variation on $\boldsymbol{\mu}$ and $\boldsymbol{\theta}$ in terms of their respective variation budget $B_{\boldsymbol{\theta}}$ and $B_{\boldsymbol{\mu}}$, and define the total variation budget $B$ as the summation of these two variation budgets:

$$B_{\boldsymbol{\theta}} = \sum_{k=2}^{K} \sum_{h=1}^{H} \|\boldsymbol{\theta}_{h,k} - \boldsymbol{\theta}_{h,k-1}\|_2, \quad B_{\boldsymbol{\mu}} = \sum_{k=2}^{K} \sum_{h=1}^{H} \|\boldsymbol{\mu}_{h,k}(\mathcal{S}) - \boldsymbol{\mu}_{h,k-1}(\mathcal{S})\|_2, \quad B = B_{\boldsymbol{\theta}} + B_{\boldsymbol{\mu}},$$

where $\boldsymbol{\mu}_{h,k}(\mathcal{S})$ is the concatenation of $\boldsymbol{\mu}_{h,k}(s)$ for all states.

**Remark 1.** *One may find the definition of $B_{\boldsymbol{\mu}}$ to be restrictive, since $B_{\boldsymbol{\mu}}$ sums over all possible states and the number of states can be infinite in linear MDPs. However, as indicated by Theorem 1, we cannot remove the dependency of $B_{\boldsymbol{\mu}}$ in the worst case. It is possible to define a nonstationarity measure that does not depend on all states if we impose additional constraints on linear MDPs. For example, we can add some reachability constraints on the state space to avoid the dependency on all states to some extent. But such reaching probability in general would also change in the nonstationary environment; therefore, it is hard to define and capture this. Future work may define a new environment change measure.*

## 3 Minimax Regret Lower Bound

In this section, we derive minimax regret lower bounds for nonstationary linear MDPs in both inhomogeneous and homogeneous settings, which quantify the fundamental difficulty when measured by the dynamic regret

in nonstationary linear MDPs. More specifically, we consider inhomogeneous setting in this paper, where the transition function $P_h^k$ (as introduced in Section 1) can be different for different $h$. In contrast, for the homogeneous setting, the transition function $P_h^k$ will be the same within an episode, i.e., for any $k$, $P_h^k \equiv P^k$ for any $h = \{1, \ldots, H\}$. All of the detailed proofs for this section are in Appendix A.

For the homogeneous setting, the minimax dynamic regret lower bound is the following.

**Theorem 1.** *For any algorithm, the dynamic regret is at least $\Omega(B^{1/3}d^{2/3}H^{1/3}T^{2/3})$ for one nonstationary homogeneous linear MDP instance, if $d \geq 4$, $T \geq 64(d-3)^2 H$.*

*Sketch of Proof.* The construction of the lower bound instance is based on the construction used in Theorem 8 in Appendix A. Nature divides the whole time horizon into $\lceil \frac{K}{N} \rceil$ intervals of equal length $N$ episodes (the last episode may be shorter). For each interval, nature initiates a new stationary linear MDP parametrized by $\boldsymbol{v}$, which is drawn from a set $\{\pm\sqrt{d-3}/\sqrt{N}\}^{d-3}$. Note that nature chooses the parameters for the linear MDP for each interval only depending on the learner's policy, and the worst-case regret for each interval is at least $\Omega(d\sqrt{H^2 N})$. Since there are at least $\lceil \frac{K}{N} \rceil - 1$ intervals, the total regret is at least $\Omega(d\sqrt{H^2 K^2 N^{-1/2}})$. By checking the total variation budget $B$, we can obtain the lower bound for $N$, which is $\Omega(B^{-2/3}d^{2/3}K^{2/3})$. Then we can obtain the desired regret lower bound. For the detailed proof, please refer to Appendix A. $\square$

For the inhomogeneous setting, the minimax dynamic regret lower bound is the following.

**Theorem 2.** *For any algorithm, the dynamic regret is at least $\Omega(B^{1/3}d^{5/6}HT^{2/3})$ for one nonstationary inhomogenous linear MDP instance, if $d \geq 4$, $H \geq 3$, $T \geq (d-1)^2 H^2/2$.*

*Sketch of Proof.* The proof idea is similar to that of Theorem 1. The only difference is that within each piecewise-stationary segment, we use the hard instance constructed by Zhou et al. (2021); Hu et al. (2022) for inhomogenous linear MDPs. Optimizing the length of each piecewise-stationary segment $N$ and the variation magnitude between consecutive segments (subject to the constraints of the total variation budget) leads to our lower bound. $\square$

Comparing these lower bounds, we can see that learning in inhomogeneous settings is indeed harder since there is more freedom in the transition function.

## 4 `LSVI-UCB-Restart` **Algorithm**

In this section, we describe our proposed algorithm `LSVI-UCB-Restart`, and discuss how to tune the hyper-parameters for cases when local variation is known or unknown. For both cases, we present their respective regret bounds. Detailed proofs are deferred to Appendix B. Note that our algorithms are all designed for inhomogeneous setting.

### 4.1 Algorithm Description

Our proposed algorithm `LSVI-UCB-Restart` has two key ingredients: least-squares value iteration with upper confidence bound to properly handle the exploration-exploitation trade-off (Jin et al., 2020), and restart strategy to adapt to the unknown nonstationarity. Our algorithm is summarized in Algorithm 1. From a high-level point of view, our algorithm runs in epochs. At each epoch, we first estimate the action-value function by solving a regularized least-squares problem from historical data, then construct the upper confidence bound for the action-value function, and update the policy greedily w.r.t. action-value function plus the upper confidence bound. Finally, we periodically restart our algorithm to adapt to the nonstationary nature of the environment.

Next, we delve into more details of the algorithm design. Note that in a linear MDP, for any policy $\pi$, the $Q$-function is linear in the feature embedding $\phi(\cdot, \cdot)$. As a preliminary, we briefly introduce least-squares value iteration (Bradtke & Barto, 1996; Osband et al., 2016), which is the key tool to estimate $\boldsymbol{w}$, the latent vector used to form the optimal action-value function, $Q_{h,k}^*(\cdot, \cdot) = \langle \phi(\cdot, \cdot), \boldsymbol{w} \rangle$. Least-squares value iteration

is a natural extension of classical value iteration algorithm (Sutton & Barto, 2018), which finds the optimal action-value function by recursively applying Bellman optimality equation,

$$Q_{h,k}^*(s,a) = [r_{h,k} + \mathbb{P}_h^k \max_{a' \in \mathcal{A}} Q_{h+1,k-1}^*(\cdot, a')](s,a).$$

In practice, the transition function $\mathbb{P}$ is unknown, and the state space might be so large that it is impossible for the learner to fully explore all states. If we parametrize the action-value function in a linear form as $\langle \phi(\cdot, \cdot), \boldsymbol{w} \rangle$, it is natural to solve a regularized least-squares problems using collected data inspired by classical value iteration. Specifically, the update formula of $\boldsymbol{w}_h^k$ in Algorithm 1 (line 8) is the analytic solution of the following regularized least-squares problem:

$$\boldsymbol{w}_h^k = \arg\min_{\boldsymbol{w}} \sum_{l=\tau}^{k-1} [r_{h,l}(s_h^l, a_h^l) + \max_{a \in \mathcal{A}} Q_{h+1}^{k-1}(s_{h+1}^l, a) - \langle \phi(s_h^l, a_h^l), \boldsymbol{w} \rangle]^2 + \|\boldsymbol{w}\|_2.$$

One might be skeptical since simply applying least-squares method to solve $\boldsymbol{w}$ does not take the distribution drift in $\mathbb{P}$ and $r$ into account and hence, may lead to non-trivial estimation error. However, we show that the estimation error can gracefully adapt to the nonstationarity, and it suffices to restart the estimation periodically to achieve good dynamic regret.

In addition to least-squares value iteration, the inner loop of Algorithm 1 also adds an additional quadratic bonus term $\beta \|\phi(\cdot, \cdot)\|_{(\Lambda_h^k)^{-1}}$ (line 9) to encourage exploration, where $\beta$ is a scalar and $\Lambda_h^k$ is the Gram matrix of the regularized least-squares problem. Intuitively, $1/\|\phi\|_{(\Lambda_h^k)^{-1}}^{-1}$ is the effective sample number of the agent observed so far in the direction of $\phi$, thus the quadratic bonus term can quantify the uncertainty of estimation. We will show later if we tune $\beta_k$ properly, then our action-value function estimate $Q_h^k$ can be an optimistic upper bound or an approximately optimistic upper bound of the optimal action-value function, so we can adapt the principle of optimism in the face of uncertainty (Auer et al., 2002a) to explore.

Finally, we use epoch restart strategy to adapt to the drifting environment, which achieves near-optimal dynamic regret notwithstanding its simplicity. Specifically, we restart the estimation of $\boldsymbol{w}$ after $\frac{W}{H}$ episodes, all illustrated in the outer loop of Algorithm 1. Note that in general epoch size $W$ can vary for different epochs, but we find that a fixed length is sufficient to achieve near-optimal performance.

To sum up, our algorithm follows the algorithmic principles of `LSVI-UCB` (Jin et al., 2020). The key difference and challenge is to set the confidence parameter properly and use periodic restart strategy to handle nonstationarity.

---

**Algorithm 1** `LSVI-UCB-Restart` Algorithm

---

**Require:** time horizon $T$, epoch size $W$
 1: Set epoch counter $j = 1$.
 2: **while** $j \leq \lceil \frac{T}{W} \rceil$ **do**
 3:   set $\tau = (j-1)\frac{W}{H}$
 4:   **for all** $k = \tau, \tau+1, \ldots, \min(\tau + \frac{W}{H} - 1, K)$ **do**
 5:     Receive the initial state $s_1^k$.
 6:     **for all** step $h = H, \ldots, 1$ **do**
 7:       $\Lambda_h^k \leftarrow \sum_{l=\tau}^{k-1} \phi(s_h^l, a_h^l) \phi(s_h^l, a_h^l)^\top + I$
 8:       $\boldsymbol{w}_h^k \leftarrow (\Lambda_h^k)^{-1} \sum_{l=\tau}^{k-1} \phi(s_h^l, a_h^l)[r_{h,l}(s_h^l, a_h^l) + \max_a Q_{h+1}^{k-1}(s_{h+1}^l, a)]$
 9:       $Q_h^k(\cdot, \cdot) \leftarrow \min\{(\boldsymbol{w}_h^k)^\top \phi(\cdot, \cdot) + \beta_k \|\phi(\cdot, \cdot)\|_{(\Lambda_h^k)^{-1}}, H\}$
10:     **end for**
11:     **for all** step $h = 1, \ldots, H$ **do**
12:       take action $a_h^k \leftarrow \arg\max_a Q_h^k(s_h^k, a)$, and observe $s_{h+1}^k$
13:     **end for**
14:   **end for**
15:   set $j = j + 1$
16: **end while**

---

### 4.2 Regret Analysis

Now we derive the dynamic regret bounds for `LSVI-UCB-Restart`, first introducing additional notation for local variations. We let

$$B_{\boldsymbol{\theta},\mathcal{E}} = \sum_{k \in \mathcal{E}} \sum_{h=1}^{H} \|\boldsymbol{\theta}_{h,k} - \boldsymbol{\theta}_{h,k-1}\|_2 \text{ and } B_{\boldsymbol{\mu},\mathcal{E}} = \sum_{k \in \mathcal{E}} \sum_{h=1}^{H} \|\boldsymbol{\mu}_{h,k}(\mathcal{S}) - \boldsymbol{\mu}_{h,k-1}(\mathcal{S})\|_2 \tag{2}$$

be the local variation for $\boldsymbol{\theta}$ and $\boldsymbol{\mu}$ in epoch $\mathcal{E}$. To derive the dynamic regret lower bounds, we need the following lemma to control the fluctuation of least-squares value iteration.

**Lemma 1.** *(Modified from Jin et al. (2020)) Denote $\tau$ to be the first episode in the epoch which contains episode $k$. There exists an absolute constant $C$ such that the following event $E$,*

$$\left\|\sum_{l=\tau}^{k-1} \boldsymbol{\phi}_h^l [V_{h+1}^k(s_{h+1}^l) - \mathbb{P}_h^l V_{h+1}^k(s_h^l, a_h^l)]\right\|_{(\Lambda_h^k)^{-1}} \leq CdH\sqrt{\log[2(c_\beta+1)dW/p]}, \quad \forall (k,h) \in \mathcal{E} \times [H].$$

*happens with probability at least $1 - p/2$.*

Next we proceed to derive the dynamic regret bounds for two cases: (1) local variations are known, and (2) local variations are unknown.

#### 4.2.1 Known Local Variations

For the case of known local variations, under event $E$ defined in Lemma 1, the estimation error of least-squares estimation scales with the quadratic bonus $\|\boldsymbol{\phi}(\cdot,\cdot)\|_{(\Lambda_h^k)^{-1}}$ uniformly for any policy $\pi$, which is detailed in the following lemma.

**Lemma 2.** *Under event $E$ defined in Lemma 1, we have for any policy $\pi$, $\forall s, a, h, k \in \mathcal{S} \times \mathcal{A} \times [H] \times \mathcal{E}$,*

$$|\langle \boldsymbol{\phi}(s,a), \boldsymbol{w}_h^k \rangle - Q_{h,k}^\pi(s,a) - \mathbb{P}_h^k(V_{h+1}^k - V_{h+1,k}^\pi)(s,a)| \leq \beta_k \|\boldsymbol{\phi}(s,a)\|_{(\Lambda_h^k)^{-1}},$$

*where $\beta_k = C_0 dH\sqrt{\log(2dW/p)} + B_{\boldsymbol{\theta},\mathcal{E}}\sqrt{d(k-\tau)} + B_{\boldsymbol{\mu},\mathcal{E}}H\sqrt{d(k-\tau)}$ and $\tau$ is the first episode in the current epoch.*

The proof of Lemma 2 is included in Appendix B. Based on Lemma 2, we can show that if we set $\beta_k$ properly with knowledge of local variations $B_{\boldsymbol{\theta},\mathcal{E}}$ and $B_{\boldsymbol{\mu},\mathcal{E}}$, the action-value function estimate maintained in Algorithm 1 is an upper bound of the optimal action-value function under event $E$.

**Lemma 3.** *Under event $E$ defined in Lemma 1, for episode $k$, if we set $\beta_k = cdH\sqrt{\log(2dW/p)} + B_{\boldsymbol{\theta},\mathcal{E}}\sqrt{d(k-\tau)} + B_{\boldsymbol{\mu},\mathcal{E}}H\sqrt{d(k-\tau)}$, we have*

$$Q_h^k(s,a) \geq Q_{h,k}^*, \quad \forall (s,a,h,k) \in \mathcal{S} \times \mathcal{A} \times [H] \times \mathcal{E}.$$

*Sketch of Proof.* For the last step $H$ in each episode, the results hold due to Lemma 2 since after step $H$ there is no reward and the episode terminates. We then prove $Q_h^k$ is indeed the upper bound for the optimal action-value function $Q_{h,k}^*$ for remaining $h \in [H-1]$ by induction. Please see Appendix B for details. □

After showing the action-value function estimate is the optimistic upper bound of the optimal action-value function, we can derive the dynamic regret bound within one epoch via recursive regret decomposition. The dynamic regret within one epoch for Algorithm 1 with the knowledge of $B_{\boldsymbol{\theta},\mathcal{E}}$ and $B_{\boldsymbol{\mu},\mathcal{E}}$ is as follows, and the proof is deferred to Appendix B.

**Theorem 3.** *For each epoch $\mathcal{E}$ with epoch size $W$, set $\beta$ in the $k$-th episode as $\beta_k = cdH\sqrt{\log(2dW/p)} + B_{\boldsymbol{\theta},\mathcal{E}}\sqrt{d(k-\tau)} + B_{\boldsymbol{\mu},\mathcal{E}}H\sqrt{d(k-\tau)}$, where $c$ is an absolute constant and $p \in (0,1)$. Then the dynamic regret within that epoch is $\tilde{O}(H^{3/2}d^{3/2}W^{1/2} + B_{\boldsymbol{\theta},\mathcal{E}}dW + B_{\boldsymbol{\mu},\mathcal{E}}dHW)$ with probability at least $1 - p$.*

By summing over all epochs and applying the union bound, we can obtain the dynamic regret upper bound for `LSVI-UCB-Restart` for the whole time horizon.

**Theorem 4.** *If we set $\beta_k = cdH\sqrt{\log(2dT/p)} + B_{\theta,\mathcal{E}}\sqrt{d(k-\tau)} + B_{\mu,\mathcal{E}}H\sqrt{d(k-\tau)}$, the dynamic regret of* `LSVI-UCB-Restart` *is $\tilde{O}(H^{3/2}d^{3/2}TW^{-1/2} + B_{\theta}dW + B_{\mu}dHW)$, with probability at least $1-p$.*

By properly tuning the epoch size $W$, we can obtain a tight dynamic regret upper bound.

**Corollary 1.** *Let $W = \lceil B^{-2/3}T^{2/3}d^{1/3}H^{-2/3} \rceil H$, and $\beta_k = cdH\sqrt{\log(2dW/p)} + B_{\theta,\mathcal{E}}\sqrt{d(k-\tau)} + B_{\mu,\mathcal{E}}H\sqrt{d(k-\tau)}$ for each epoch.* `LSVI-UCB-Restart` *achieves $\tilde{O}(B^{1/3}d^{4/3}H^{4/3}T^{2/3})$ dynamic regret, with probability at least $1-p$.*

**Remark 2.** *Corollary 1 shows that if local variations are known, we can achieve near-optimal dependency on the the total variation $B_{\theta}, B_{\mu}$ and time horizon $T$ compared to the lower bound provided in Theorem 1. However, the dependency on $d$ and $H$ is worse. The dependency on $d$ is unlikely to improve unless there is an improvement to LSVI-UCB.*

**Remark 3.** *The definition of total variation $B$ is related to the misspecification error defined by Jin et al. (2020). One can apply the Cauchy-Schwarz inequality to show that our total variation bound implies that misspecification in Eq. (4) of Jin et al. is also bounded (but not vice versa). However, the regret analysis in the misspecified linear MDP of Jin et al. (2020) is restricted to static regret, so we cannot directly borrow their analysis for the misspecified setting (Jin et al., 2020) to handle our dynamic regret (as defined in Eq. (1)).*

**Remark 4.** *For the case when the environment changes abruptly $L$ times, our algorithm enjoys an $\tilde{O}(L^{1/3}T^{2/3})$ dynamic regret bound, which is sub-optimal compared to Wei & Luo (2021). The reason is that periodic restart is not a suitable strategy to handle abrupt changes since the passive nature indicates that we cannot guarantee detecting the abrupt environment change within a reasonably short delay. Wei & Luo (2021) overcome this issue by running two tests on top of multiple base instances with different scales to detect the environmental change. Similar ideas have also been used in the piecewise-stationary bandit literature (Besson & Kaufmann, 2019), where a change detection subroutine is run to detect the environmental change, so the regret incurred by the environmental drift can be better controlled.*

### 4.2.2 Unknown Local Variation

If the local variations are unknown, the proof is similar to the case of known local variations. We only highlight the differences compared to the previous case. The key difference is that without knowledge of local variations $B_{\theta}$ and $B_{\mu}$, we set the hyper-parameter $\beta = cdH\sqrt{\log(2dW/p)}$. As a result, the action-value function estimate $Q_h^k$ maintained in Algorithm 1 is no longer the optimistic upper bound of the optimal action-value function, but only approximately, up to some error, proportional to the local variation. The rigorous statement is as follows.

**Lemma 4.** *Under event $E$ defined in Lemma 1, if we set $\beta = cdH\sqrt{\log(2dW/p)}$, we have*

$$\forall (s,a,h,k) \in \mathcal{S} \times \mathcal{A} \times [H] \times \mathcal{E},$$
$$Q_h^k(s,a) \geq Q_{h,k}^* - (H-h+1)(B_{\theta,\mathcal{E}}\sqrt{d(k-\tau)} + B_{\mu,\mathcal{E}}H\sqrt{d(k-\tau)}).$$

*Sketch of Proof.* For the case when local variations are unknown, the least -squares estimation error have some additional terms that are linear in the local variations $B_{\theta,\mathcal{E}}$ and $B_{\mu,\mathcal{E}}$ (See Lemma 10 in Appendix B). Then we can prove that $Q_h^k$ is an approximate upper bound of $Q_{h,k}^*$ via induction. For details, please see Appendix B. $\square$

By applying a similar proof technique as Theorem 3, we can derive the dynamic regret within one epoch when local variations are unknown.

**Theorem 5.** *For each epoch $\mathcal{E}$ with epoch size $W$, if we set $\beta_k = cdH\sqrt{\log(2dW/p)}$, where $c$ is an absolute constant and $p \in (0,1)$, then the dynamic regret within that epoch is $\tilde{O}(\sqrt{d^3H^3W} + B_{\theta,\mathcal{E}}\sqrt{d/H}W^{3/2} + B_{\mu,\mathcal{E}}\sqrt{dH}W^{3/2})$ with probability at least $1-p$, where $B_{\theta,\mathcal{E}}$ and $B_{\mu,\mathcal{E}}$ are the total variation within that epoch.*

By summing regret over epochs and applying a union bound over all epochs, we obtain the dynamic regret of `LSVI-UCB-Restart` for the whole time horizon.

**Theorem 6.** *If we set $\beta = cdH\sqrt{\log(2dT/p)}$, then the dynamic regret of `LSVI-UCB-Restart` is $\tilde{O}(d^{3/2}H^{3/2}TW^{-1/2} + B_{\boldsymbol{\theta}}d^{1/2}H^{-1/2}W^{3/2} + B_{\boldsymbol{\mu}}d^{1/2}H^{1/2}W^{3/2})$, with probability at least $1 - p$.*

By properly tuning the epoch size $W$, we can obtain a tight regret bound for the case of unknown local variations as follows.

**Corollary 2.** *Let $W = \lceil B^{-1/2}T^{1/2}d^{1/2}H^{-1/2} \rceil H$ and $\beta_k = cdH\sqrt{\log(2dW/p)}$. Then `LSVI-UCB-Restart` achieves $\tilde{O}(B^{1/4}d^{5/4}H^{5/4}T^{3/4})$ dynamic regret, with probability at least $1 - p$.*

**Remark 5.** *Our algorithm has a slightly worse regret bound compared with Wei & Luo (2021). However, our algorithm has a much better better computational complexity, since we only require to maintain one instance of the base algorithm. We are also the first one to conduct numerical experiments on online exploration for non-stationary MDPs (Section 6). How to achieve the $\tilde{O}(T^{2/3})$ dynamic regret bound without prior knowledge and with only one base instance is still an open problem.*

## 5 `Ada-LSVI-UCB-Restart`: a Parameter-free Algorithm

In practice, the total variations $B_{\boldsymbol{\theta}}$ and $B_{\boldsymbol{\mu}}$ are unknown. To mitigate this issue, we present a parameter-free algorithm `Ada-LSVI-UCB-Restart` and its dynamic regret bound.

### 5.1 Algorithm Description

Inspired by bandit-over-bandit mechanism (Cheung et al., 2019), we develop a new parameter-free algorithm. The key idea is to use `LSVI-UCB-Restart` as a subroutine (set $\beta = cdH\sqrt{\log(2dT/p)}$ since we assume total variations are unknown), and periodically update the epoch size based on the historical data under the time-varying $\mathbb{P}$ and $r$ (potentially adversarial). More specifically, `Ada-LSVI-UCB-Restart` (Algorithm 2) divides the whole time horizon into $\lceil \frac{T}{HM} \rceil$ blocks of equal length $M$ episodes (the length of the last block can be smaller than $M$ episodes), and specifies a set $J_W$ from which epoch size is drawn. For each block $i \in [\lceil \frac{T}{HM} \rceil]$, `Ada-LSVI-UCB` runs a master algorithm to select the epoch size $W_i$ and runs `LSVI-UCB-Restart` with $W_i$ for the current block. After the end of this block, the total reward of this block is fed back to the master algorithm, and the posteriors of the parameters are updated accordingly.

For the detailed master algorithm, we select `EXP3-P` (Bubeck & Cesa-Bianchi, 2012) since it is able to deal with non-oblivious adversary. Now we present the details of `Ada-LSVI-UCB-Restart`. We set the length of each block $M$ and the feasible set of epoch size $J_W$ as follows:

$$M = \lceil 5T^{1/2}d^{1/2}H^{-1/2} \rceil, J_W = \{H, 2H, 4H, \ldots, MH\}.$$

The intuition of designing the feasible set for epoch size $J_W$ is to guarantee it can well-approximate the optimal epoch size with the knowledge of total variations while on the other hand make it as small as possible, so the learner do not lose much by adaptively selecting the epoch size from $J_W$. This intuition is more clear when we derive the dynamic regret bound of `Ada-LSVI-UCB-Restart`. Denoting $|J_W| = \Delta$, the master algorithm `EXP3-P` treats each element of $J_W$ as an arm and updates the probabilities of selecting each feasible epoch size based on the reward collected in the past. It begins by initializing

$$\alpha = 0.95\sqrt{\frac{\ln \Delta}{\Delta \lceil T/MH \rceil}}, \ \beta = \sqrt{\frac{\ln \Delta}{\Delta \lceil T/MH \rceil}}, \ \gamma = 1.05\sqrt{\frac{\ln \Delta}{\Delta \lceil T/MH \rceil}}, \ q_{l,1} = 0, \ l \in [\Delta], \quad (3)$$

where $\alpha, \beta, \gamma$ are parameters used in `EXP3-P` and $q_{l,1}, l \in [\Delta]$ are the initialization of the estimated total reward of running different epoch lengths. At the beginning of the block $i$, the agent first sees the initial state $s_1^{(i-1)H}$, and updates the probability of selecting different epoch lengths for block $i$ as

$$u_{l,i} = (1 - \gamma)\frac{\exp(\alpha q_{l,i})}{\sum_{l \in [\Delta]} \exp(\alpha q_{l,i})} + \frac{\gamma}{\Delta}. \quad (4)$$

Then the master algorithm samples $l_i \in [\Delta]$ according to the updated distribution $\{u_{l,i}\}_{i\in[\Delta]}$; the epoch size $W_i$ for the block $i$ is chosen as $l_i$-th element in $J_W$, $\lfloor M^{l_i/\lfloor \ln M \rfloor} \rfloor H$. After selecting the epoch size $W_i$, $\texttt{Ada-LSVI-UCB}$ runs a new copy of $\texttt{LSVI-UCB-Restart}$ with that epoch size. By the end of each block, $\texttt{Ada-LSVI-UCB-Restart}$ observes the total reward of the current block, denoted as $R_i(W_i, s_1^{(i-1)H})$, then the algorithm updates the estimated total reward of running different epoch sizes (divide $R_i(W_i, s_1^{(i-1)H})$ by $MH$ to normalize):

$$q_{l,i+1} = q_{l,i} + \frac{\beta + \mathbb{1}\{l = l_i\} R_i(W_i, s_1^{(i-1)H})/MH}{u_{l,i}}. \tag{5}$$

---

**Algorithm 2** $\texttt{ADA-LSVI-UCB-Restart}$ Algorithm

---

**Require:** time horizon $T$, block length $M$, feasible set of epoch size $J_w$
1: Initialize $\alpha$, $\beta$, $\gamma$ and $\{q_{l,1}\}_{l\in[\Delta]}$ according to Eq. 3.
2: **for all** $i = 1, 2, \ldots, \lceil T/HM \rceil$ **do**
3:     Receive the initial state $s_1^{(i-1)H}$
4:     Update the epoch size selection distribution $\{u_{l,i}\}_{l\in[\Delta]}$ according to Eq. 4
5:     Sample $l_i \in [\Delta]$ from the updated distribution $\{u_{l,i}\}_{l\in[\Delta]}$, then set the epoch size for block $i$ as $W_i = \lfloor M^{l_i/\lfloor \ln M \rfloor} \rfloor H$.
6:     **for all** $t = (i-1)MH + 1, \ldots, \min(iMH, T)$ **do**
7:       Run $\texttt{LSVI-UCB-Restart}$ algorithm with epoch size $W_i$
8:     **end for**
9:     After observing the total reward for block $i$, $R_i(W_i, s_1^{(i-1)H})$, update the estimated total reward of running different epoch sizes $\{q_{l,i+1}\}_{l\in[\Delta]}$ according to Eq. 5
10: **end for**

---

### 5.2 Regret Analysis

Now we present the dynamic regret bound achieved by $\texttt{Ada-LSVI-UCB-Restart}$.

**Theorem 7.** *The dynamic regret of $\texttt{Ada-LSVI-UCB-Restart}$ is $\tilde{O}(B^{1/4}d^{5/4}H^{5/4}T^{3/4})$.*

*Sketch of Proof.* To analyze the dynamic regret of $\texttt{Ada-LSVI-UCB-Restart}$, we decompose the dynamic regret into two terms. The first term is regret incurred by always selecting the best $W^\dagger$ from $J_W$, and the second term is regret incurred by adaptively selecting the epoch size from $J_W$ via $\texttt{EXP3-P}$ rather than always selecting $W^\dagger$. For the second term, we reduce to an adversarial bandit problem and directly use the regret bound of $\texttt{EXP3-P}$ (Bubeck & Cesa-Bianchi, 2012; Auer et al., 2002b). For the first term, we show that $W^\dagger$ can well-approximate the optimal epoch size with the knowledge of $B_{\boldsymbol{\mu}}$ and $B_{\boldsymbol{\theta}}$, up to constant factors. Thus we can use Theorem 6 to bound the first term. For details, see Appendix C. $\qquad\square$

**Remark 6.** *Using the master algorithm to select the window size is reminiscent of model selection approaches to online RL (Agarwal et al., 2017; Pacchiano et al., 2020; Lee et al., 2021; Abbasi-Yadkori et al., 2020). Typically model selection approaches achieve worse rates compared to the best base algorithm. However, in our setting we can discretize the feasible set for epoch size $J_W$ at a proper granularity, so we can control the additional regret incurred by the master algorithm with respect to the best epoch size in $J_W$ to be reasonably small. That is, we have more freedom to choose the base algorithms compared to some problem settings in the model selection literature. As a result, the regret bound does not lose much compared to the case where we have knowledge of total variation $B$.*

## 6 Experiments

In this section, we perform empirical experiments on synthetic datasets to illustrate the effectiveness of $\texttt{LSVI-UCB-Restart}$ and $\texttt{Ada-LSVI-UCB-Restart}$. We compare the cumulative rewards of the proposed algorithms with five baseline algorithms: $\texttt{Epsilon-Greedy}$ (Watkins, 1989), $\texttt{Random-Exploration}$,

LSVI-UCB (Jin et al., 2020), OPT-WLSVI (Touati & Vincent, 2020), and MASTER (Wei & Luo, 2021). As discussed before, we are the first one to perform numerical experiments on online exploration for non-stationary MDPs and demonstrate the effectiveness of proposed algorithms.

The agent takes actions uniformly in Random-Exploration. In Epsilon-Greedy, instead of adding a bonus term as in LSVI-UCB, the agent takes the greedy action according to the current estimate of $Q$ function with probability $1 - \epsilon$, and takes the action uniformly at random with probability $\epsilon$, where we set $\epsilon = 0.05$. For LSVI-UCB and LSVI-UCB-Restart, we set $\beta = 0.001cdH\sqrt{\log(200dT)}$. In addition, for LSVI-UCB-Restart we test the performance of two cases: (1) known global variation, where we set $W = \lceil B^{-1/2}T^{1/2}d^{1/2}H^{-1/2}\rceil H$; (2) unknown global variation (denoted LSVI-UCB-Unknown), where we set $W = \lceil T^{1/2}d^{1/2}H^{-1/2}\rceil H$ (the dynamic regret bound is $\tilde{O}(Bd^{5/4}H^{5/4}T^{3/4})$ for this case). For ADA-LSVI-UCB-Restart, we set the length of each block $M = \lceil 0.2T^{1/2}d^{1/2}H^{1/2}\rceil$. Note that the tuning of hyperparameters is different from our theoretical derivations by some constant factors. The reason is that the worst-case analysis is pessimistic and we ignore the constant factor in the derivation.

**Settings** We consider an MDP with $S = 15$ states, $A = 7$ actions, $H = 10$, $d = 10$, and $T = 20000$. In the *abruptly-changing* environment, the linear MDP changes abruptly every 100 episodes to another linear MDP with different transition function and reward function. The changes happen periodically by cycling through different linear MDPs; we have 5 different linear MDPs in total. In the gradually-changing environment, we consider the same set of 5 linear MDPs, $\{M_0, M_1, \ldots, M_4\}$. The environment changes smoothly from $M_i$ to $M_{(i+1) \mod 5}$ over every 100 episodes. To be more specific, at episode $100i$ ($i$ is a non-negative integer), the MDP model is $M_{i \mod 5}$ parameterized by latent vectors $\{\boldsymbol{\theta}_h^{i \mod 5}\}_{h=1}^H$ and $\{\boldsymbol{\mu}_h^{i \mod 5}(\mathcal{S})\}_{h=1}^H$. The latent vectors of the MDP from episode $100i$ to $100(i+1)$ are the linear interpolations of those of $M_{i \mod 5}$ and $M_{(i+1) \mod 5}$, i.e., at episode $k$ ($100i \leq k \leq 100(i+1)$), $\boldsymbol{\theta}_{h,k} = (1-\frac{k-100i}{100})\boldsymbol{\theta}_h^{i \mod 5} + \frac{k-100i}{100}\boldsymbol{\theta}_h^{(i+1) \mod 5}, \forall h \in [H]$, $\boldsymbol{\mu}_{h,k}(\mathcal{S}) = (1 - \frac{k-100i}{100})\boldsymbol{\mu}_h^{i \mod 5}(\mathcal{S}) + \frac{k-100i}{100}\boldsymbol{\mu}_h^{(i+1) \mod 5}(\mathcal{S}), \forall h \in [H]$. To make the environment challenging for exploration, our construction falls into the category of combination lock (Koenig & Simmons, 1993). For each of these 5 linear MDPs, there is only one good (and different) chain that contains a huge reward at the end, but 0 reward for the rest of the chain. Further, any sub-optimal action has small positive rewards that would attract the agent to depart from the optimal route. Therefore, the agent must perform "deep exploration" (Osband et al., 2019) to obtain near-optimal policy. The details of the constructions are in Appendix E. Here we report the cumulative rewards and the running time of all algorithms averaged over 10 trials.

From Figure 1, we see LSVI-UCB-Restart with the knowledge of global variation drastically outperforms all other methods designed for stationary environments , in both abruptly-changing and gradually-changing environments, since it restarts the estimation of the $Q$ function with knowledge of the total variations. Ada-LSVI-UCB-Restart also outperforms the baselines because it also takes the nonstationarity into account by periodically updating the epoch size for restart. In addition, Ada-LSVI-UCB-Restart has a huge gain compared to LSVI-UCB-Unknown, which agrees with our theoretical analysis. This suggests that Ada-LSVI-UCB-Restart works well when the knowledge of global variation is unavailable. Our proposed algorithms not only perform systemic exploration, but also adapt to the environment change.

Compared to OPT-WLSVI and MASTER, our proposed algorithms achieve comparable empirical performance. More specifically, MASTER outperforms our proposed algorithm which agrees with its dynamic regret upper bound. However, the variance of MASTER is larger due to the random scheduling of multiple base algorithms. Our algorithm outperforms OPT-WLSVI in the abrupt change setting, but has worse performance in the gradual change setting, which agrees with the empirical findings in the nonstationary contextual bandit literature (Zhao et al., 2020). The main advantage of our algorithm compared to OPT-WLSVI and MASTER is its computational efficiency, as demonstrated by Figure 1. The reason is that our algorithm only requires the most recent data to estimate the $Q$-function, while the other two require the entire history. MASTER additionally requires maintaining multiple base instances at different scales, which further increases the computational burden.

From Figure 1, we find that the restart strategy works better under abrupt changes than under gradual changes, since the gap between our algorithms and the baseline algorithms designed for stationary environ-

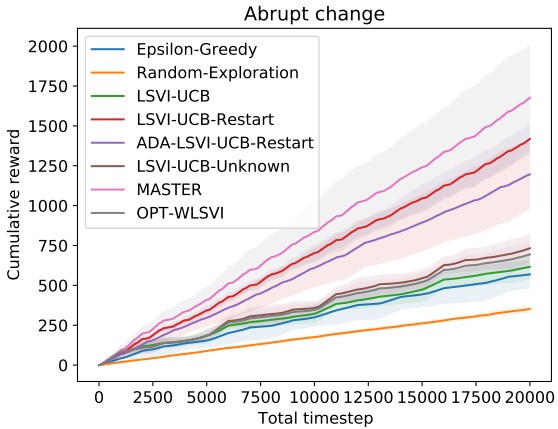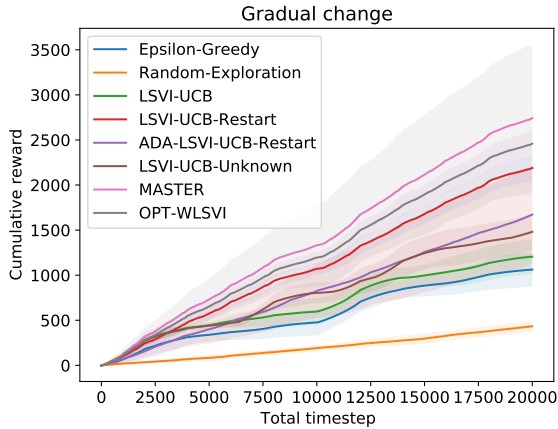

Figure 1: Comparisons of different methods on cumulative reward under two different environments. The results are averaged over 10 trials and the error bars show the standard deviations. The environment changes abruptly in the left subfigure, whereas the environment changes gradually in the right subfigure.

ments is larger in this setting. The reason is that the algorithms designed to explore in stationary MDPs are generally insensitive to abrupt change in the environment. For example, UCB-type exploration does not have incentive to take actions other than the one with the largest upper confidence bound of $Q$-value, and if it has collected sufficient number of samples, it very likely never explores the new optimal action thereby taking the former optimal action forever. On the other hand, in gradually-changing environment, `LSVI-UCB` and `Epsilon-Greedy` can perform well in the beginning when the drift of environment is small. However, when the change of environment is greater, they no longer yield satisfactory performance since their $Q$ function estimate is quite off. This also explains why `LSVI-UCB` and `Epsilon-Greedy` outperform `ADA-LSVI-UCB` at the beginning in the gradually-changing environment, as shown in Figure 1.

Figure 2 shows that the running times of `LSVI-UCB-Restart` and `Ada-LSVI-UCB-Restart` are roughly the same. They are much less compared with `MASTER`, `OPT-WLSVI`, `LSVI-UCB`, `Epsilon-Greedy`. This is because `LSVI-UCB-Restart` and `Ada-LSVI-UCB-Restart` can automatically restart according to the variation of the environment and thus have much smaller computational burden since it does not need to use the entire history to compute the current policy at each time step. The running time of `LSVI-UCB-Unknown` is larger than `LSVI-UCB-restart` since the epoch larger is larger due to the lack of the knowlege of total variation $B$, but it still does not use the entire history to compute its policy. Although `Random-Exploration` takes the least time, it cannot find the near-optimal policy. This result further demonstrates that our algorithms are not only sample-efficient, but also computationally tractable.

# 7 Conclusion and Future Work

In this paper, we studied nonstationary RL with time-varying reward and transition functions. We focused on the class of nonstationary linear MDPs such that linear function approximation is sufficient to realize any value function. We first incorporated the epoch start strategy into `LSVI-UCB` algorithm (Jin et al., 2020) to propose the `LSVI-UCB-Restart` algorithm with low dynamic regret when the total variations are known. We then designed a parameter-free algorithm `Ada-LSVI-UCB-Restart` that enjoys a slightly worse dynamic regret bound without knowing the total variations. We derived a minimax regret lower bound for nonstationary linear MDPs to demonstrate that our proposed algorithms are near-optimal. Specifically, when the local variations are known, `LSVI-UCB-Restart` is near order-optimal except for the dependency on feature dimension $d$, planning horizon $H$, and some poly-logarithmic factors. Numerical experiments demonstrates the effectiveness of our algorithms.

A number of future directions are of interest. An immediate step is to investigate whether the dependence on the dimension $d$ and planning horizon $H$ in our bounds can be improved, and whether the minimax regret

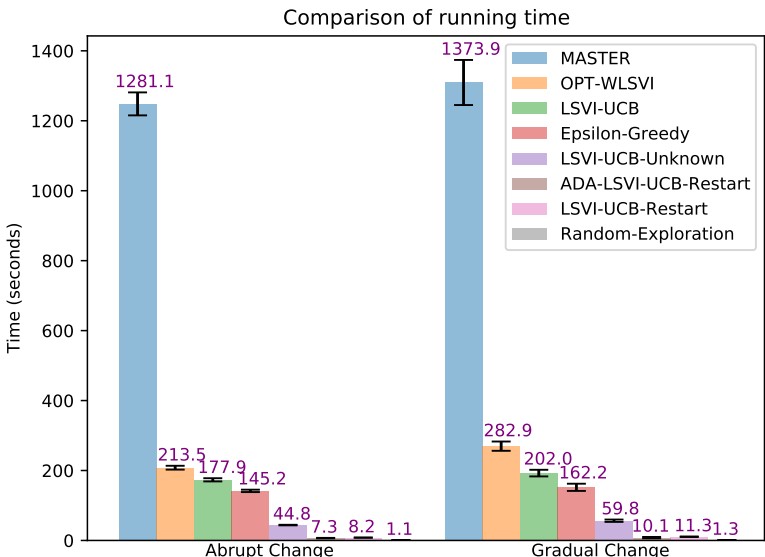

Figure 2: Comparisons of different methods on running time for two different environments. The results are averaged over 10 trials and the error bars show the standard deviations. See Appendix E.2 for details on hardware.

lower bound can also be improved. It would also be interesting to investigate the setting of nonstationary RL under general function approximation (Wang et al., 2020; Du et al., 2021; Jin et al., 2021), which is closer to modern RL algorithms in practice. Recall that our algorithm is more computationally efficient than other works. Another closely related and interesting direction is to study the low-switching cost (Gao et al., 2021) or deployment efficient (Huang et al., 2021) algorithm in the nonstationary RL setting. Finally, our algorithm is based on the Optimism in Face of Uncertainty. There is another broad category of algorithms called Thompson Sampling (TS) (Agrawal & Jia, 2017; Russo, 2019; Agrawal et al., 2021; Xiong et al., 2021; Ishfaq et al., 2021; Dann et al., 2021; Zhang, 2022). It would be an interesting avenue to see whether empirically appealing TS algorithms are also suitable in nonstationary RL settings.

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

## A    Proofs in Section 3

To prove the minimax dynamic regret lower bound for the homogeneous setting, we first prove the minimax regret bound for linear MDP with homogeneous transition function.

**Theorem 8.** *For any algorithm, if $d \geq 4$ and $T \geq 64(d-3)^2 H$, then there exists at least one stationary linear MDP instance that incurs regret at least $\Omega(d\sqrt{HT})$.*

The key step of this proof is to construct the hard-to-learn MDP instances. Inspired by the lower bound construction for stochastic contextual bandits (Dani et al., 2008; Lattimore & Szepesvári, 2020), we construct an ensemble of hard-to-learn 3-state linear MDPs, which is illustrated in Figure 3. This construction can be viewed as a generalization of the lower bound construction for linear contextual bandits (Dani et al., 2008; Lattimore & Szepesvári, 2020). The intuition is that the reward distributions under optimal and suboptimal policies for these instances are close: thus it is statistically hard for any learner to identify the optimal policy.

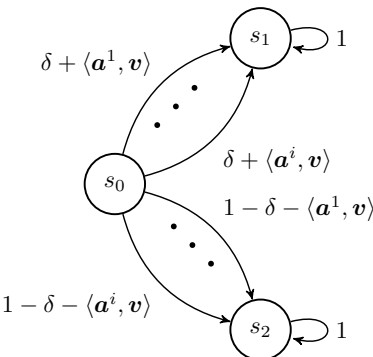

Figure 3: Graphical illustration of the hard-to-learn linear MDP instances with deterministic reward.

Each linear MDP instance in this ensemble has three states $s_0, s_1, s_2$ ($s_1$ and $s_2$ are absorbing states), and it is characterized by a unique $(d-3)$-dimensional vector $\{\pm\sqrt{(d-3)H}/\sqrt{T}\}^{d-3}$. Specifically, the vector $\boldsymbol{v}$ defines the transition function of the corresponding MDP, as illustrated in Figure 3. Each action $a$ of this MDP instance is encoded by a $(d-3)$ dimensional vector $\boldsymbol{a} \in \left\{\pm 1/\sqrt{d-3}\right\}^{d-3}$. The reward functions for the three states are fixed regardless of the actions, specifically, $r(s_0, a) = r(s_2, a) = 0, r(s_1, a) = 1, \forall a \in \mathcal{A}$. For each episode, the agent starts at $s_0$, and ends at step $H$. The transition functions of the linear MDP parametrized by $\boldsymbol{v}$ are defined as follows,

$$\mathbb{P}(s_1|s_0, a) = \delta + \langle \boldsymbol{a}, \boldsymbol{v}\rangle, \ \mathbb{P}(s_2|s_0, a) = 1 - \delta - \langle \boldsymbol{a}, \boldsymbol{v}\rangle, \ \mathbb{P}(s_1|s_1, a) = 1, \ \mathbb{P}(s_2|s_2, a) = 1,$$

where $\delta = \frac{1}{4}$. Notice that the optimal policy for the MDP instance parametrized by $\boldsymbol{v}$ is taking the action that maximizes the probability to reach $s_1$, which is equivalent to taking the action such that its corresponding vector $\boldsymbol{a}$ satisfies $\text{sgn}(\boldsymbol{a}_i) = \text{sgn}(\boldsymbol{v}_i), \forall i \in [d-3]$. Furthermore, it can be verified that the above MDP instance is indeed a linear MDP, by setting:

$$\boldsymbol{\phi}(s_0, a) = (0, 1, \delta, \boldsymbol{a}), \boldsymbol{\phi}(s_1, a) = (1, 0, 0, \vec{0}), \boldsymbol{\phi}(s_2, a) = (0, 1, 0, \vec{0})$$

$$\boldsymbol{\mu}(s_0) = (0, 0, 0, \vec{0}), \boldsymbol{\mu}(s_1) = (1, 0, 1, \boldsymbol{v}), \boldsymbol{\mu}(s_2) = (0, 1, -1, -\boldsymbol{v}), \boldsymbol{\theta} = (1, 0, 0, 0).$$

**Remark 7.** *Note that the above parameters violate the normalization assumption in Definition 1, but it is straightforward to normalize them. We ignore the additional rescaling to clarify the presentation.*

After constructing the ensemble of hard instances, we can derive the minimax regret lower bound for stationary linear MDP for the homogeneous setting.

*Proof of Theorem 8.* Let $\mathbb{P}_{t,\boldsymbol{v}}^{\pi}$ (assume $t$ is a multiple of $H$) be the probability distribution of $\{a_1^1, \sum_{h=1}^{H} r_h^1, a_1^2, \sum_{h=1}^{H} r_h^2, \ldots, a_1^{t/H}, \sum_{h=1}^{H} r_h^{t/H}\}$ of running algorithm $\pi$ on linear MDP parametrized by $\boldsymbol{v}$. First note that by the Markov property of $\pi$, we can decompose $D_{KL}(\mathbb{P}_{t,\boldsymbol{v}}^{\pi}||\mathbb{P}_{t,\boldsymbol{v'}}^{\pi})$ as

$$\sum_{l=1}^{t/H} \mathbb{E} D_{KL}\left(\mathbb{P}\left(\sum_{h=1}^{H} r_h^l|a_1^l, \boldsymbol{v}\right) || \mathbb{P}\left(\sum_{h=1}^{H} r_h^l|a_1^l, \boldsymbol{v'}\right)\right).$$

Recall that due to our hard cases construction, the first step in every episode determines the distribution of the total reward of that episode, thus

$$D_{KL}\left(\mathbb{P}\left(\sum_{h=1}^{H} r_h^l|a_1^l, \boldsymbol{v}\right) || \mathbb{P}\left(\sum_{h=1}^{H} r_h^l|a_1^l, \boldsymbol{v'}\right)\right)$$
$$= \left(\delta + \langle \boldsymbol{a}_1^l, \boldsymbol{v}\rangle\right) \log \frac{\delta + \langle \boldsymbol{a}_1^l, \boldsymbol{v}\rangle}{\delta + \langle \boldsymbol{a}_1^l, \boldsymbol{v'}\rangle} + \left(1 - \delta - \langle \boldsymbol{a}_1^l, \boldsymbol{v}\rangle\right) \log \frac{1 - \delta - \langle \boldsymbol{a}_1^l, \boldsymbol{v}\rangle}{1 - \delta - \langle \boldsymbol{a}_1^l, \boldsymbol{v'}\rangle} \quad (6)$$

We bound the KL divergence in (6) applying the following lemma.

**Lemma 5.** *(Auer et al., 2010) If $0 \leq \delta' \leq 1/2$ and $\epsilon' \leq 1 - 2\delta'$, then*

$$\delta' \log \frac{\delta'}{\delta' + \epsilon'} + (1 - \delta') \log \frac{1 - \delta'}{1 - \delta' - \epsilon'} \leq \frac{2(\epsilon')^2}{\delta'}.$$

To apply Lemma 5, we let $\langle \boldsymbol{a}_1^l, \boldsymbol{v}\rangle + \delta = \delta'$, $\langle \boldsymbol{v} - \boldsymbol{v'}, \boldsymbol{a}_1^l\rangle = \epsilon'$. Thus we must ensure the following inequalities hold for any $\boldsymbol{a}, \boldsymbol{v}, \boldsymbol{v'}$:

$$\langle \boldsymbol{a}, \boldsymbol{v}\rangle + \delta \leq \frac{(d-3)\sqrt{H}}{\sqrt{T}} + \delta \leq 1/2$$
$$\langle \boldsymbol{v} - \boldsymbol{v'}, \boldsymbol{a}\rangle \leq \frac{2(d-3)\sqrt{H}}{\sqrt{T}} \leq 1 - 2\left(\frac{(d-3)\sqrt{H}}{\sqrt{T}} + \delta\right) \leq 1 - 2\delta'.$$

To guarantee the above inequalities hold, we can set $\delta = \frac{1}{4}$ and let $\frac{(d-3)\sqrt{H}}{\sqrt{T}} \leq \frac{1}{8}$. Now we get back to bounding Eq. 6. Let $\Delta = \frac{(d-3)\sqrt{H}}{\sqrt{T}}$ and suppose $\boldsymbol{v}$ and $\boldsymbol{v'}$ only differ in one coordinate. Then

$$D_{KL}\left(\mathbb{P}\left(\sum_{h=1}^{H} r_h^l|a_1^l, \boldsymbol{v}\right) || \mathbb{P}\left(\sum_{h=1}^{H} r_h^l|a_1^l, \boldsymbol{v'}\right)\right) \leq \frac{8\Delta^2 \frac{1}{(d-3)^2}}{\delta - \Delta} \leq \frac{16\Delta^2}{\delta(d-3)^2} \leq \frac{64H}{T}.$$

Furthermore, let $E_{i,b}$ be the the following event:

$$|\{l \in [K] : \text{sgn}(\boldsymbol{a}_1^l)_i \neq \text{sgn}(b)\}| \geq \frac{1}{2}K.$$

Let $q_{i,\boldsymbol{v}} = \mathbb{P}[E_{i,v_i}|\boldsymbol{v}]$, the probability that the agent is taking sub-optimal action for the $i$-th coordinate for at least half of the episodes given that the underlying linear MDP is parameteriezed by $\boldsymbol{v}$. We can then lower bound the regret of any algorithm when running on linear MDP parameterized by $\boldsymbol{v}$ as:

$$\text{Reg}_{\boldsymbol{v}}(T) \geq \sum_{i=1}^{d-3} q_{i,\boldsymbol{v}} K(H-1)\sqrt{\frac{H}{T}}$$
$$\geq \left(\sqrt{TH} - \sqrt{K}\right) \sum_{i=1}^{d-3} q_{i,\boldsymbol{v}}, \quad (7)$$

since whenever the learner takes a sub-optimal action that differs from the optimal action by one coordinate, it will incur $2\sqrt{\frac{H}{T}}(H-1)$ expected regret. Next we take the average over $2^{d-3}$ linear MDP instances to

show that on average it incurs $\Omega(d\sqrt{HT})$ regret, thus there exists at least one instance incurring $\Omega(d\sqrt{HT})$ regret. Before that, we need to bound the summation of bad events under two close linear MDP instances. Denote the vector which is only different from $\boldsymbol{v}$ in $i$-th coordinate as $\boldsymbol{v}^{\oplus i}$. Then we have

$$
\begin{aligned}
q_{i,\boldsymbol{v}} + q_{i,\boldsymbol{v}^{\oplus i}} &= \mathbb{P}[E_{i,\boldsymbol{v}_i}|\boldsymbol{v}] + \mathbb{P}[E_{i,\boldsymbol{v}_i^{\oplus i}}|\boldsymbol{v}^{\oplus i}] \\
&= \mathbb{P}[E_{i,\boldsymbol{v}_i}|\boldsymbol{v}] + \mathbb{P}[\bar{E}_{i,\boldsymbol{v}_i}|\boldsymbol{v}^{\oplus i}] \\
&\geq \frac{1}{2}\exp(-D_{KL}(P_{T,\boldsymbol{v}}||P_{T,\boldsymbol{v}^{\oplus i}})) \\
&\geq \frac{1}{2}\exp(-64),
\end{aligned} \tag{8}
$$

where the inequality is due to Bretagnolle-Huber inequality (Bretagnolle & Huber, 1979). Now we are ready to lower bound the average regret over all linear MDP instances.

$$
\begin{aligned}
\frac{1}{2^{d-3}}\sum_{\boldsymbol{v}}\mathrm{Reg}_{\boldsymbol{v}}(T) &\geq \frac{\sqrt{HT}-\sqrt{K}}{2^{d-3}}\sum_{\boldsymbol{v}}\sum_{i=1}^{d-3}q_{i,\boldsymbol{v}} \\
&\geq \frac{\sqrt{HT}-\sqrt{K}}{2^{d-3}}\sum_{i=1}^{d-3}\sum_{\boldsymbol{v}}\frac{q_{i,\boldsymbol{v}}+q_{i,\boldsymbol{v}^{\oplus i}}}{2} \\
&\geq \frac{\sqrt{HT}-\sqrt{K}}{2^{d-3}}2^{d-3}\frac{1}{4}e^{-64}(d-3) \\
&\gtrsim \Omega(d\sqrt{HT})
\end{aligned}
$$

where the first inequality is due to (7), and the third inequality is due to (8). $\qquad\square$

Based on Theorem 8, we can derive the minimax dynamic regret for nonstationary linear MDP.

*Proof of Theorem 1.* We construct the hard instance as follows: We first divide the whole time horizon $T$ into $\lceil\frac{K}{N}\rceil$ intervals, where each interval has $\lceil\frac{K}{N}\rceil$ episodes (the last interval might be shorter if $K$ is not a multiple of $N$). For each interval, the linear MDP is fixed and parameterized by a $\boldsymbol{v} \in \{\pm\frac{\sqrt{(d-3)}}{\sqrt{N}}\}^{d-3}$ which we define when constructing the hard instances in Theorem 8. Note that different intervals are completely decoupled, thus information is not passed across intervals. For each interval, it incurs regret at least $\Omega(d\sqrt{H^2N})$ by Theorem 8. Thus the total regret is at least

$$
\begin{aligned}
\mathrm{Dyn\text{-}Reg}(T) &\gtrsim (\lceil\frac{K}{N}\rceil - 1)\Omega(d\sqrt{H^2N}) \\
&\gtrsim \Omega(d\sqrt{H^2K^2}N^{-1/2}). 
\end{aligned} \tag{9}
$$

Intuitively, we would like $N$ to be as small as possible to obtain a tight lower bound. However, due to our construction, the total variation for two consecutive blocks is upper-bounded by

$$
\sqrt{\sum_{i=1}^{d-3}\frac{4(d-3)}{N}} = \frac{2(d-3)}{\sqrt{N}}.
$$

Note that the total time variation for the whole time horizon is $B$ and by definition $B \geq \frac{2(d-3)}{\sqrt{N}}(\lfloor\frac{K}{N}\rfloor - 1)$, which implies $N \gtrsim \Omega(B^{-2/3}d^{2/3}K^{2/3})$. Substituting the lower bound of $N$ into (9), we have

$$
\mathrm{Dyn\text{-}Reg}(T) \gtrsim \Omega(B^{1/3}d^{2/3}K^{2/3}H) \gtrsim \Omega(B^{1/3}d^{2/3}H^{1/3}T^{2/3})
$$

which concludes the proof. $\qquad\square$

Finally, we provide the formal proof for Theorem 2.

*Proof of Theorem 2.* We construct the hard instance as follows. We first divide the whole time horizon $T$ into $\lceil \frac{K}{N} \rceil$ intervals, where each interval has $\lceil \frac{K}{N} \rceil$ episodes (the last interval might be shorter if $K$ is not a multiple of $N$). For each interval, the linear MDP is fixed and parameterized by a $\boldsymbol{v} \in \{\pm \frac{1}{4\sqrt{2}} \sqrt{\frac{1}{NH}}\}^{d-1}$, defined in Lemma E.1 in Hu et al. (2022). Note that different intervals are completely decoupled, thus information is not passed across intervals. For each interval, it incurs regret at least $\Omega(d\sqrt{H^3 N})$ by Lemma E.1 in Hu et al. (2022). Thus the total regret is at least

$$\text{Dyn-Reg}(T) \gtrsim (\lceil \frac{K}{N} \rceil - 1)\Omega(d\sqrt{H^3 N})$$
$$\gtrsim \Omega(d\sqrt{H^3 K^2}N^{-1/2}). \tag{10}$$

Intuitively, we would like $N$ to be as small as possible to obtain a tight lower bound. However, due to our construction, the total variation for two consecutive blocks is upper-bounded by

$$\sqrt{\sum_{i=1}^{d-1} \frac{1}{32NH}} = \frac{\sqrt{d-1}}{4\sqrt{2}\sqrt{NH}}.$$

Note that the total time variation for the whole time horizon is $B$ and by definition $B \gtrsim \Omega(d^{1/2}KN^{-3/2}H^{-1/2})$, which implies $N \gtrsim \Omega(B^{-2/3}d^{1/3}K^{2/3}H^{-1/3})$. Substituting the lower bound of $N$ into Eq. (10), we have

$$\text{Dyn-Reg}(T) \gtrsim \Omega(B^{1/3}d^{5/6}HT^{2/3})$$

which concludes the proof. □

## B  Proofs in Section 4

Here we provide the proofs in Section 4. We first want to comment that our algorithm builds on `LSVI-UCB`. `LSVI` can be seen as a specialization of the regression-based Fitted Q-Iteration algorithm (Ernst et al., 2005; Munos & Szepesvári, 2008; Chen & Jiang, 2019) to the linear case, and `LSVI-UCB` (Jin et al., 2020) further adds the bonus term on top of that to handle exploration.

Now, we introduce some notations we use throughout the proof. We let $\boldsymbol{w}_h^k$, $\Lambda_h^k$ and $Q_h^k$ as the parameters and action-value function estimate in episode $k$ for step $h$. Denote value function estimate as $V_h^k(s) = \max_a Q_h^k(s, a)$. For any policy $\pi$, we let $\boldsymbol{w}_{h,k}^\pi$, $Q_{h,k}^\pi$ be the ground-truth parameter and action-value function for that policy in episode $k$ for step $h$. We also abbreviate $\boldsymbol{\phi}(s_h^l, a_h^l)$ as $\boldsymbol{\phi}_h^l$ for notational simplicity.

We first work on the case when local variation is known and then consider the case when local variation is unknown.

### B.1  Case 1: Known Local Variation

Before we prove the regret upper bound within one epoch (Theorem 5), we need some additional lemmas. The first lemma is used to control the fluctuations in least-squares value iteration, when performed on the value function estimate $V_h^k(\cdot)$ maintained in Algorithm 1.

*Proof of Lemma 1.* The lemma is slightly different than Jin et al. (2020, Lemma B.3), since they assume $\mathbb{P}_h$ is fixed for different episodes. It can be verified that the proof for stationary case still holds in our case without any modifications since the results in Jin et al. (2020) holds for least-squares value iteration for arbitrary function in the function class of our interest, i.e., $\{V | V = \{\boldsymbol{\phi}(\cdot, \cdot), \boldsymbol{w}\}, \boldsymbol{w} \in \mathbb{R}^d\}$. Let us first restate Lemma B.3 in Jin et al. (2020) to compare against our Lemma 1.

**Lemma 6.** *(Lemma B.3 in Jin et al. (2020)) There exists an absolute constant $C$ that is independent of $c_\beta$ such that for any fixed $p \in [0, 1]$, if we let the event $E$ be the following,*

$$\forall (k, h) \in [K] \times [H], \quad \left\| \sum_{l=1}^{k-1} \phi_h^l [V_{h+1}^k(s_{h+1}^l) - \mathbb{P}_h V_{h+1}^k(s_h^l, a_h^l)] \right\|_{(\Lambda_h^k)^{-1}} \leq CdH\sqrt{\log[2(c_\beta + 1)dW/p]},$$

*then event $E$ happens with probability at least $1 - p/2$.*

To prove Lemma 1, we need the following technical lemmas.

**Lemma 7.** *(Lemma D.4 in Jin et al. (2020)) Let $\{x_\tau\}_{\tau=1}^\infty$ be a stochastic process on state space $\mathcal{S}$ with corresponding filtration $\{\mathcal{F}_\tau\}_{\tau=1}^\infty$. Let $\{\phi_\tau\}_{\tau=1}^\infty$ be a $\mathbb{R}^d$-valued stochastic process where $\phi_\tau \in \mathcal{F}_{\tau-1}$, and $\|\phi_\tau\| \leq 1$. Let $\Lambda_k = \lambda I + \sum_{\tau=1}^k \phi_\tau \phi_\tau^\top$. Then for any $\delta > 0$, with probability at least $1 - \delta$, for all $k > 0$, and $V \in \mathcal{V}$ so that $\sup_x |V(x)| \leq H$, we have*

$$\left\| \sum_{\tau=1}^k \phi_\tau \{V(x_\tau) - \mathbb{E}[V(x_\tau)|\mathcal{F}_{\tau-1}]\} \right\|_{\Lambda_k^{-1}}^2 \leq 4H^2 [\frac{d}{2} \log(\frac{k+\lambda}{\lambda}) + \log \frac{\mathcal{N}_\epsilon}{\delta}] + \frac{8k^2\epsilon^2}{\lambda},$$

*where $\mathcal{N}_\epsilon$ is the covering number for $\mathcal{V}$.*

**Lemma 8.** *(Lemma D.6 in Jin et al. (2020)) For function class with the following form,*

$$\mathcal{V}(\cdot) = \min\{\max_a \boldsymbol{w}^\top \boldsymbol{\phi}(\cdot, a) + \beta \sqrt{\boldsymbol{\phi}(\cdot, a)^\top \Lambda^{-1} \boldsymbol{\phi}(\cdot, a)}, H\}$$

*satisfying the constraints given by Jin et al. (2020), the log covering number $\log \mathcal{N}_\epsilon$ for this function class is upper bounded by $d \log(1 + 8H\sqrt{d}/\epsilon) + d^2 \log[1 + 8d^{1/2}\beta^2/(\lambda\epsilon^2)]$.*

By applying Lemma 7 and Lemma 8, and setting $\lambda = 1, \epsilon = dH/k, \beta = CdH \log(2dT/p)$, we prove Lemma 1.
$\square$

We then proceed to derive the error bound for the action-value function estimate maintained in the algorithm for any policy.

*Proof of Lemma 2.* Note that $Q_{h,k}^\pi(s, a) = \langle \phi(s, a), \boldsymbol{w}_{h,k}^\pi \rangle$. First we can decompose $\boldsymbol{w}_h^k - \boldsymbol{w}_{h,k}^\pi$ as

$$\boldsymbol{w}_h^k - \boldsymbol{w}_{h,k}^\pi = (\Lambda_h^k)^{-1} \sum_{l=\tau}^{k-1} \phi_h^l [r_h^l(s_h^l, a_h^l) + V_{h+1}^k(s_{h+1}^l)] - \boldsymbol{w}_{h,k}^\pi$$

$$= (\Lambda_h^k)^{-1} \{ -\boldsymbol{w}_{h,k}^\pi + \sum_{l=\tau}^{k-1} \phi_h^l [V_{h+1}^k(s_{h+1}^l) - \mathbb{P}_h^k V_{h+1,k}^\pi(s_h^l, a_h^l)] + \sum_{l=\tau}^{k-1} \phi_h^l [r_h^l(s_h^l, a_h^l) - r_h^k(s_h^l, a_h^l)] \}$$

$$= \underbrace{-(\Lambda_h^k)^{-1} \boldsymbol{w}_{h,k}^\pi}_{①} + \underbrace{(\Lambda_h^k)^{-1} \sum_{l=\tau}^{k-1} \phi_h^l [V_{h+1}^k(s_{h+1}^l) - \mathbb{P}_h^l V_{h+1}^k(s_h^l, a_h^l)]}_{②}$$

$$+ \underbrace{(\Lambda_h^k)^{-1} \sum_{l=\tau}^{k-1} \phi_h^l [(\mathbb{P}_h^l - \mathbb{P}_h^k) V_{h+1}^k(s_h^l, a_h^l)]}_{③} + \underbrace{(\Lambda_h^k)^{-1} \sum_{l=\tau}^{k-1} \phi_h^l \mathbb{P}_h^k (V_{h+1}^k - V_{h+1,k}^\pi)(s_h^l, a_h^l)}_{④}$$

$$+ \underbrace{(\Lambda_h^k)^{-1} \sum_{l=\tau}^{k-1} \phi_h^l [r_h^l(s_h^l, a_h^l) - r_h^k(s_h^l, a_h^l)]}_{⑤}.$$

We bound the individual terms on right side one by one. For the first term,

$$
\begin{aligned}
|\langle \boldsymbol{\phi}(s,a), \text{①}\rangle| &= |\langle \boldsymbol{\phi}(s,a), (\Lambda_h^k)^{-1} \boldsymbol{w}_{h,k}^\pi\rangle| \\
&\leq \left\| \boldsymbol{w}_{h,k}^\pi \right\| \left\| \boldsymbol{\phi}(s,a)\right\|_{(\Lambda_h^k)^{-1}} \\
&\leq 2H\sqrt{d}\left\| \boldsymbol{\phi}(s,a)\right\|_{(\Lambda_h^k)^{-1}},
\end{aligned}
$$

where the last inequality is due to Lemma 12. For the second term, we know that under event $E$ defined in Lemma 1,

$$
|\langle \boldsymbol{\phi}(s,a), \text{②}\rangle| \leq CdH\sqrt{\log[2(c_\beta+1)dW/p]}\left\| \boldsymbol{\phi}(s,a)\right\|_{(\Lambda_h^k)^{-1}}.
$$

For the third term,

$$
\begin{aligned}
\langle \boldsymbol{\phi}(s,a), \text{③}\rangle &= \langle \boldsymbol{\phi}(s,a), (\Lambda_h^k)^{-1} \sum_{l=\tau}^{k-1} \boldsymbol{\phi}_h^l [(\mathbb{P}_h^l - \mathbb{P}_h^k)V_{h+1}^k(s_h^l, a_h^l)]\rangle \\
&\leq \sum_{l=\tau}^{k-1} |\boldsymbol{\phi}(s,a)^\top (\Lambda_h^k)^{-1} \boldsymbol{\phi}_h^l| |[(\mathbb{P}_h^l - \mathbb{P}_h^k)V_{h+1}^k(s_h^l, a_h^l)] \\
&\leq B_{\boldsymbol{\mu}, \mathcal{E}} H \sum_{l=\tau}^{k-1} |\boldsymbol{\phi}(s,a)^\top (\Lambda_h^k)^{-1} \boldsymbol{\phi}_h^l| \\
&\leq B_{\boldsymbol{\mu}, \mathcal{E}} H \sqrt{\sum_{l=\tau}^{k-1} \|\boldsymbol{\phi}(s,a)\|_{(\Lambda_h^k)^{-1}}^2} \sqrt{\sum_{l=\tau}^{k-1} (\boldsymbol{\phi}_h^l)^\top (\Lambda_h^k)^{-1} \boldsymbol{\phi}_h^l} \\
&\leq \sqrt{d(k-\tau)} B_{\boldsymbol{\mu}, \mathcal{E}} H \left\| \boldsymbol{\phi}(s,a)\right\|_{(\Lambda_h^k)^{-1}},
\end{aligned}
$$

where the first three inequalities are due to Cauchy-Schwarz inequality and boundedness of $\mathbb{P}_h^l - \mathbb{P}_h^k$ and $V_{h+1}^k$, and the last inequality is due to Lemma 13.

For the fourth term,

$$
\begin{aligned}
\langle \boldsymbol{\phi}(s,a), \text{④}\rangle &= \langle \boldsymbol{\phi}(s,a), (\Lambda_h^k)^{-1} \sum_{l=\tau}^{k-1} \boldsymbol{\phi}_h^l \mathbb{P}_h^k (V_{h+1}^k - V_{h+1,k}^\pi)(s_h^l, a_h^l)\rangle \\
&= \langle \boldsymbol{\phi}(s,a), (\Lambda_h^k)^{-1} \sum_{l=\tau}^{k-1} \boldsymbol{\phi}_h^l (\boldsymbol{\phi}_h^l)^\top \int (V_{h+1}^k(s') - V_{h+1,k}^\pi(s'))d\boldsymbol{\mu}_{h,k}(s')\rangle \\
&= \underbrace{\langle \boldsymbol{\phi}(s,a), \int (V_{h+1}^k - V_{h+1,k}^\pi)(s')d\boldsymbol{\mu}_{h,k}(s')\rangle}_{\text{⑥}} - \underbrace{\langle \boldsymbol{\phi}(s,a), (\Lambda_h^k)^{-1} \int (V_{h+1}^k - V_{h+1,k}^\pi)(s')d\boldsymbol{\mu}_{h,k}(s')\rangle}_{\text{⑦}},
\end{aligned}
$$

where $\text{⑥} = [\mathbb{P}_h^k(V_{h+1}^k - V_{h+1,k}^\pi)](s,a)$ and $\text{⑦} \leq 2H\sqrt{d}\left\| \boldsymbol{\phi}(s,a)\right\|_{(\Lambda_h^k)^{-1}}$ due to Cauchy-Schwarz inequality.

For the fifth term,

$$
\begin{aligned}
\langle \boldsymbol{\phi}(s,a), \text{⑤}\rangle &= \langle \boldsymbol{\phi}(s,a), (\Lambda_h^k)^{-1} \sum_{l=\tau}^{k-1} \boldsymbol{\phi}_h^l [r_h^l(s_h^l, a_h^l) - r_h^k(s_h^l, a_h^l)]\rangle \\
&\leq \sum_{l=\tau}^{k-1} |\boldsymbol{\phi}(s,a)^\top (\Lambda_h^k)^{-1} \boldsymbol{\phi}_h^l| |r_h^l(s_h^l, a_h^l) - r_h^k(s_h^l, a_h^l)| \\
&\leq \sqrt{d(k-\tau)} B_{\boldsymbol{\theta}, \mathcal{E}} \left\| \boldsymbol{\phi}(s,a)\right\|_{(\Lambda_h^k)^{-1}},
\end{aligned}
$$

where the inequalities are derived similarly as bounding the third term. After combining all the upper bounds for these individual terms, we have

$$
\begin{aligned}
&|\langle \boldsymbol{\phi}(s,a), \boldsymbol{w}_h^k \rangle - Q_{h,k}^\pi(s,a) - \mathbb{P}_h^k \left( V_{h+1}^k - V_{h+1,k}^\pi \right)(s,a)| \\
&\leq 4H\sqrt{d}\, \|\boldsymbol{\phi}(s,a)\|_{(\Lambda_h^k)^{-1}} + CdH\sqrt{\log[2(c_\beta+1)dW/p]}\, \|\boldsymbol{\phi}(s,a)\|_{(\Lambda_h^k)^{-1}} \\
&\quad + B_{\boldsymbol{\theta},\mathcal{E}}\sqrt{d(k-\tau)}\, \|\boldsymbol{\phi}(s,a)\|_{(\Lambda_h^k)^{-1}} + B_{\boldsymbol{\mu},\mathcal{E}}H\sqrt{d(k-\tau)}\, \|\boldsymbol{\phi}(s,a)\|_{(\Lambda_h^k)^{-1}} \\
&\leq C_0 dH\sqrt{\log[2dW/p]}\, \|\boldsymbol{\phi}(s,a)\|_{(\Lambda_h^k)^{-1}} + B_{\boldsymbol{\theta},\mathcal{E}}\sqrt{d(k-\tau)}\, \|\boldsymbol{\phi}(s,a)\|_{(\Lambda_h^k)^{-1}} \\
&\quad + B_{\boldsymbol{\mu},\mathcal{E}}H\sqrt{d(k-\tau)}\, \|\boldsymbol{\phi}(s,a)\|_{(\Lambda_h^k)^{-1}} .
\end{aligned}
$$

The second inequality holds if we choose a sufficiently large absolute constant $C_0$. □

Lemma 2 implies that the action-value function estimate we maintained in Algorithm 1 is always an optimistic upper bound of the optimal action-value function with high confidence, if we know the local variation.

*Proof of Lemma 3.* We prove this by induction. First prove the base case when $h = H$. According to Lemma 2, we have

$$
|\langle \boldsymbol{\phi}(s,a), \boldsymbol{w}_H^k \rangle - Q_{H,k}^*(s,a)| \leq \beta_k \|\boldsymbol{\phi}(s,a)\|_{(\Lambda_H^k)^{-1}} ,
$$

which implies

$$
Q_H^k(s,a) = \min\{\langle \boldsymbol{w}_H^k, \boldsymbol{\phi}(s,a) \rangle + \beta_k \|\boldsymbol{\phi}(s,a)\|_{(\Lambda_H^k)^{-1}} , H\} \geq Q_{H,k}^*(s,a).
$$

Now suppose the statement holds true at step $h+1$, then for step $h$, due to Lemma 2, we have

$$
|\langle \boldsymbol{\phi}(s,a), \boldsymbol{w}_h^k \rangle - Q_{h,k}^\pi(s,a) - \mathbb{P}_h^k(V_{h+1}^k - V_{h+1,k}^*)(s,a)| \leq \beta_k \|\boldsymbol{\phi}(s,a)\|_{(\Lambda_h^k)^{-1}} .
$$

By the induction hypothesis, we have $\mathbb{P}_h^k(V_{h+1}^k - V_{h+1,k}^*)(s,a) \geq 0$, thus

$$
Q_h^k(s,a) = \min\{\langle \boldsymbol{w}_h^k, \boldsymbol{\phi}(s,a) \rangle + \beta_k \|\boldsymbol{\phi}(s,a)\|_{(\Lambda_H^k)^{-1}} , H\} \geq Q_{h,k}^*(s,a).
$$

□

Next we derive the bound for the gap between the value function estimate and the ground-truth value function for the executing policy $\pi^k$, $\delta_h^k = V_h^k(s_h^k) - V_{h,k}^{\pi^k}(s_h^k)$, in a recursive manner.

**Lemma 9.** *Let* $\delta_h^k = V_h^k(s_h^k) - V_{h,k}^{\pi^k}(s_h^k)$, $\zeta_{h+1}^k = \mathbb{E}[\delta_{h+1}^k | s_h^k, a_h^k] - \delta_{h+1}^k$. *Under event $E$ defined in Lemma 1, we have for all* $(k, h) \in \mathcal{E} \times [H]$,

$$
\delta_h^k \leq \delta_{h+1}^k + \zeta_{h+1}^k + 2\beta_k \|\boldsymbol{\phi}_h^k\|_{(\Lambda_h^k)^{-1}} .
$$

*Proof.* By Lemma 2, for any $(s, a, h, k) \in \mathcal{S} \times \mathcal{A} \times [H] \times \mathcal{E}$,

$$
Q_h^k(s,a) - Q_h^{\pi^k}(s,a) \leq \mathbb{P}_h^k(V_{h+1}^k - V_{h+1,k}^{\pi^k})(s,a) + 2\beta_k \|\boldsymbol{\phi}(s,a)\|_{(\Lambda_H^k)^{-1}} .
$$

Note that $Q_h^k(s_h^k, a_h^k) = \max_a Q_h^k(s_h^k, a) = V_h^k(s_h^k)$ according to Algorithm 1, and $Q_{h,k}^{\pi^k}(s_h^k, a_h^k) = V_{h,k}^{\pi^k}(s_h^k)$ by the definition. Thus,

$$
\delta_h^k \leq \delta_{h+1}^k + \zeta_{h+1}^k + 2\beta_k \|\boldsymbol{\phi}_h^k\|_{(\Lambda_h^k)^{-1}} .
$$

□

Now we are ready to derive the regret bound within one epoch.

*Proof of Theorem 3.* We denote the dynamic regret within that epoch as Dyn-Reg($\mathcal{E}$). We define $\delta_h^k = V_h^k(s_h^k) - V_{h,k}^{\pi^k}(s_h^k)$ and $\zeta_{h+1}^k = \mathbb{E}[\delta_{h+1}^k | s_h^k, a_h^k] - \delta_{h+1}^k$ as in Lemma 11. We derive the dynamic regret within a epoch $\mathcal{E}$ (the length of this epoch is $W$ which is equivalent to $\frac{W}{H}$ episodes) conditioned on the event $E$ defined in Lemma 1 which happens with probability at least $1 - p/2$.

$$
\begin{aligned}
\text{Dyn-Reg}(\mathcal{E}) &= \sum_{k \in \mathcal{E}} \left[ V_{1,k}^*(s_1^k) - V_{1,k}^{\pi^k} \right] \\
&\leq \sum_{k \in \mathcal{E}} \left[ V_1^k(s_1^k) - V_{1,k}^{\pi^k} \right] \\
&\leq \sum_{k \in \mathcal{E}} \delta_1^k \\
&\leq \sum_{k \in \mathcal{E}} \sum_{h=1}^H \zeta_h^k + 2 \sum_{k \in \mathcal{K}} \beta_k \sum_{h=1}^H \left\| \phi_h^k \right\|_{(\Lambda_h^k)^{-1}},
\end{aligned}
\tag{11}
$$

where the first inequality is due to Lemma 3, the third inequality is due to Lemma 9. For the first term in the right side, since $V_h^k$ is independent of the new observation $s_h^k$, $\{\zeta_h^k\}$ is a martingale difference sequence. Applying the Azuma-Hoeffding inequlity, we have for any $t > 0$,

$$
\mathbb{P} \left( \sum_{k \in \mathcal{E}} \sum_{h=1}^H \zeta_h^k \geq t \right) \geq \exp(-t^2/(2WH^2)).
$$

Hence with probability at least $1 - p/2$, we have

$$
\sum_{k \in \mathcal{E}} \sum_{h=1}^H \zeta_h^k \leq 2H \sqrt{W \log(2dW/p)}.
\tag{12}
$$

For the second term, we bound via Cauchy-Schwarz inequality:

$$
\begin{aligned}
2 \sum_{k \in \mathcal{E}} \beta_k \sum_{h=1}^H \left\| \phi_h^k \right\|_{(\Lambda_h^k)^{-1}} &= 2C_0 dH \sqrt{\log 2(dW/p)} \sum_{k \in \mathcal{E}} \sum_{h=1}^H \left\| \phi_h^k \right\|_{(\Lambda_h^k)^{-1}} + 2 \sum_{k \in \mathcal{E}} B_{\theta,\mathcal{E}} \sqrt{d(k-\tau)} \sum_{h=1}^H \left\| \phi_h^k \right\|_{(\Lambda_h^k)^{-1}} \\
&\quad + 2 \sum_{k \in \mathcal{E}} B_{\boldsymbol{\mu}} H \sqrt{d(k-\tau)} \sum_{h=1}^H \left\| \phi_h^k \right\|_{(\Lambda_h^k)^{-1}} \\
&\leq 2C_0 dH \sqrt{\log 2(dW/p)} \sum_{h=1}^H \sqrt{W/H} (\sum_{k \in \mathcal{E}} \left\| \phi_h^k \right\|_{(\Lambda_h^k)^{-1}}^2)^{1/2} \\
&\quad + 2 \sum_{h=1}^H (\sum_{k \in \mathcal{E}} B_{\theta,\mathcal{E}} \sqrt{d(k-\tau)})^{1/2} (\sum_{k \in \mathcal{E}} \left\| \phi_h^k \right\|_{(\Lambda_h^k)^{-1}}^2)^{1/2} \\
&\quad + 2 \sum_{h=1}^H (\sum_{k \in \mathcal{E}} B_{\boldsymbol{\mu},\mathcal{E}} H \sqrt{d(k-\tau)})^{1/2} (\sum_{k \in \mathcal{E}} \left\| \phi_h^k \right\|_{(\Lambda_h^k)^{-1}}^2)^{1/2} \\
&\leq 2C_0 dH \sqrt{\log 2(dW/p)} \sum_{h=1}^H \sqrt{W/H} (\sum_{k \in \mathcal{E}} \left\| \phi_h^k \right\|_{(\Lambda_h^k)^{-1}}^2)^{1/2} \\
&\quad + 2 \sum_{h=1}^H B_{\theta,\mathcal{E}} \sqrt{d} \frac{W}{H} (\sum_{k \in \mathcal{E}} \left\| \phi_h^k \right\|_{(\Lambda_h^k)^{-1}}^2)^{1/2} \\
&\quad + 2 \sum_{h=1}^H B_{\theta,\mathcal{E}} \sqrt{d} W (\sum_{k \in \mathcal{E}} \left\| \phi_h^k \right\|_{(\Lambda_h^k)^{-1}}^2)^{1/2}
\end{aligned}
\tag{13}
$$

By Lemma 14, we have

$$(\sum_{k \in \mathcal{E}} \|\boldsymbol{\phi}_h^k\|_{(\Lambda_h^k)^{-1}}^2)^{1/2} \le \sqrt{d \log \left( \frac{W}{H} + 1 \right)}. \tag{14}$$

Finally, by combining Eq. 11–14, we obtain the regret bound within the epoch $\mathcal{E}$ as:

$$\text{Dyn-Reg}(\mathcal{E}) \lesssim \tilde{O}(H^{3/2} d^{3/2} W^{1/2} + B_{\boldsymbol{\theta}, \mathcal{E}} dW + B_{\boldsymbol{\mu}, \mathcal{E}} dHW).$$

□

By summing over all epochs and applying a union bound, we obtain the regret bound for the whole time horizon.

**Theorem 9.** *If we set $\beta = \beta_k = cdH \sqrt{\log(2dT/p)} + B_{\boldsymbol{\theta}, \mathcal{E}} \sqrt{d(k-\tau)} + B_{\boldsymbol{\mu}, \mathcal{E}} H \sqrt{d(k-\tau)}$, the dynamic regret of `LSVI-UCB-Restart` is $\tilde{O}(H^{3/2} d^{3/2} T W^{-1/2} + B_{\boldsymbol{\theta}} dW + B_{\boldsymbol{\mu}} dHW)$, with probability at least $1 - p$.*

*Proof.* In total there are $N = \lceil \frac{T}{W} \rceil$ epochs. For each epoch $\mathcal{E}_i$ if we set $\delta = \frac{p}{N}$, then it will incur regret $\tilde{O}(d^{3/2} H^{3/2} W^{1/2} + B_{\boldsymbol{\theta}, \mathcal{E}_i} dW + B_{\boldsymbol{\mu}, \mathcal{E}_i} dHW)$ with probability at least $1 - \frac{p}{N}$. By summing over all epochs and applying the union bound over them, we can obtain the regret upper bound for the whole time horizon. With probability at least $1 - p$,

$$\begin{aligned}
\text{Dyn-Reg}(T) &= \sum_{\mathcal{E}_i} \text{Dyn-Reg}(\mathcal{E}_i) \\
&\lesssim \sum_{\mathcal{E}_i} \tilde{O}(d^{3/2} H^{3/2} W^{1/2} + B_{\boldsymbol{\theta}, \mathcal{E}_i} dW + B_{\boldsymbol{\mu}, \mathcal{E}_i} dHW) \\
&\lesssim \tilde{O}(H^{3/2} d^{3/2} T W^{-1/2} + B_{\boldsymbol{\theta}} dW + B_{\boldsymbol{\mu}} dHW).
\end{aligned}$$

□

### B.2 Case 2: Unknown Local Variation

Similar to the case of known local variation, we first derive the error bound for the action-value function estimate maintained in the algorithm for any policy, which is the following technical lemma.

**Lemma 10.** *Under event $E$ defined in Lemma 1, we have for any policy $\pi$, $\forall s, a, h, k \in \mathcal{S} \times \mathcal{A} \times [H] \times \mathcal{E}$,*

$$|\langle \boldsymbol{\phi}(s,a), \boldsymbol{w}_h^k \rangle - Q_{h,k}^\pi(s,a) - \mathbb{P}_h^k(V_{h+1}^k - V_{h+1,k}^\pi)(s,a)| \le \beta \|\boldsymbol{\phi}(s,a)\|_{(\Lambda_h^k)^{-1}} + B_{\boldsymbol{\theta}, \mathcal{E}} \sqrt{d(k-\tau)} + B_{\boldsymbol{\mu}, \mathcal{E}} H \sqrt{d(k-\tau)},$$

*where $\beta = C_0 dH \sqrt{\log(2dW/p)}$ and $\tau$ is the first episode in the current epoch.*

*Proof.* This lemma is a looser upper bound implied by Lemma 2. By Lemma 2, we have

$$\begin{aligned}
&|\langle \boldsymbol{\phi}(s,a), \boldsymbol{w}_h^k \rangle - Q_{h,k}^\pi(s,a) - \mathbb{P}_h^k(V_{h+1}^k - V_{h+1,k}^\pi)(s,a)| \\
\le &C_o dH \sqrt{\log(2dW/p)} \|\boldsymbol{\phi}(s,a)\|_{(\Lambda_h^k)^{-1}} + B_{\boldsymbol{\theta}, \mathcal{E}} \sqrt{d(k-\tau)} \|\boldsymbol{\phi}(s,a)\|_{(\Lambda_h^k)^{-1}} \\
&+ B_{\boldsymbol{\mu}, \mathcal{E}} H \sqrt{d(k-\tau)} \|\boldsymbol{\phi}(s,a)\|_{(\Lambda_h^k)^{-1}} \\
\le &C_o dH \sqrt{\log(2dW/p)} \|\boldsymbol{\phi}(s,a)\|_{(\Lambda_h^k)^{-1}} + B_{\boldsymbol{\theta}, \mathcal{E}} \sqrt{d(k-\tau)} \\
&+ B_{\boldsymbol{\mu}, \mathcal{E}} H \sqrt{d(k-\tau)},
\end{aligned}$$

where the second inequality is due to $\|\boldsymbol{\phi}(s,a)\| \le 1$ and $\lambda_{\min}(\Lambda_h^k) \ge 1$, thus $\|\boldsymbol{\phi}(s,a)\|_{(\Lambda_h^k)^{-1}} \le 1$. □

Different from Lemma 3, when the local variation is unknown, the action-value function estimate we maintained in Algorithm 1 is no longer an optimistic upper bound of the optimal action-value function, but approximately up to some error proportional to the local variation.

*Proof of Lemma 4.* We prove this by induction. First prove the base case when $h = H$. According to Lemma 10, we have

$$|\langle \boldsymbol{\phi}(s,a), \boldsymbol{w}_H^k \rangle - Q_{H,k}^*(s,a)| \leq \beta \left\| \boldsymbol{\phi}(s,a) \right\|_{(\Lambda_H^k)^{-1}} + B_{\boldsymbol{\theta},\mathcal{E}} \sqrt{d(k-\tau)} + B_{\boldsymbol{\mu},\mathcal{E}} H \sqrt{d(k-\tau)},$$

which implies

$$\begin{aligned}
Q_H^k(s,a) &= \min\{\langle \boldsymbol{w}_H^k, \boldsymbol{\phi}(s,a) \rangle + \beta \left\| \boldsymbol{\phi}(s,a) \right\|_{(\Lambda_H^k)^{-1}}, H\} \\
&\geq Q_{H,k}^*(s,a) - (B_{\boldsymbol{\theta},\mathcal{E}} \sqrt{d(k-\tau)} + B_{\boldsymbol{\mu},\mathcal{E}} H \sqrt{d(k-\tau)}).
\end{aligned}$$

Now suppose the statement holds true at step $h+1$, then for step $h$, due to Lemma 10, we have

$$\begin{aligned}
&|\langle \boldsymbol{\phi}(s,a), \boldsymbol{w}_h^k \rangle - Q_{h,k}^\pi(s,a) - \mathbb{P}_h^k(V_{h+1}^k - V_{h+1,k}^*)(s,a)| \\
\leq & \beta \left\| \boldsymbol{\phi}(s,a) \right\|_{(\Lambda_h^k)^{-1}} + B_{\boldsymbol{\theta},\mathcal{E}} \sqrt{d(k-\tau)} + B_{\boldsymbol{\mu},\mathcal{E}} H \sqrt{d(k-\tau)}.
\end{aligned}$$

By the induction hypothesis, we have $[\mathbb{P}_h^k(V_{h+1}^k - V_{h+1,k}^*)](s,a) \geq -(H - h + 2)(B_{\boldsymbol{\theta},\mathcal{E}} \sqrt{d(k-\tau)} + B_{\boldsymbol{\mu},\mathcal{E}} H \sqrt{d(k-\tau)})$, thus

$$\begin{aligned}
Q_h^k(s,a) &= \min\{\langle \boldsymbol{w}_h^k, \boldsymbol{\phi}(s,a) \rangle + \beta \left\| \boldsymbol{\phi}(s,a) \right\|_{(\Lambda_H^k)^{-1}}, H\} \\
&\geq Q_{h,k}^*(s,a) - (H - h + 1)(B_{\boldsymbol{\theta},\mathcal{E}} \sqrt{d(k-\tau)} + B_{\boldsymbol{\mu},\mathcal{E}} H \sqrt{d(k-\tau)}).
\end{aligned}$$

$\square$

Similar to Lemma 9, next we derive the bound for the gap between the value function estimate and the ground-truth value function for the executing policy $\pi^k$, $\delta_h^k = V_h^k(s_h^k) - V_{h,k}^{\pi^k}(s_h^k)$, in a recursive manner, when the local variation is unknown.

**Lemma 11.** *Let $\delta_h^k = V_h^k(s_h^k) - V_{h,k}^{\pi^k}(s_h^k)$, $\zeta_{h+1}^k = \mathbb{E}[\delta_{h+1}^k | s_h^k, a_h^k] - \delta_{h+1}^k$. Under event $E$ defined in Lemma 1, we have for all $(k,h) \in \mathcal{E} \times [H]$,*

$$\delta_h^k \leq \delta_{h+1}^k + \zeta_{h+1}^k + 2\beta \left\| \boldsymbol{\phi}_h^k \right\|_{(\Lambda_h^k)^{-1}} + B_{\boldsymbol{\theta},\mathcal{E}} \sqrt{d(k-\tau)} + B_{\boldsymbol{\mu},\mathcal{E}} H \sqrt{d(k-\tau)}.$$

*Proof.* By Lemma 10, for any $(s,a,h,k) \in \mathcal{S} \times \mathcal{A} \times [H] \times \mathcal{E}$,

$$Q_h^k(s,a) - Q_h^{\pi^k}(s,a) \leq \mathbb{P}_h^k(V_{h+1}^k - V_{h+1,k}^{\pi^k})(s,a) + 2\beta \left\| \boldsymbol{\phi}(s,a) \right\|_{(\Lambda_H^k)^{-1}} + B_{\boldsymbol{\theta},\mathcal{E}} \sqrt{d(k-\tau)} + B_{\boldsymbol{\mu},\mathcal{E}} H \sqrt{d(k-\tau)}.$$

Note that $Q_h^k(s_h^k, a_h^k) = \max_a Q_h^k(s_h^k, a) = V_h^k(s_h^k)$ according to Algorithm 1, and $Q_{h,k}^{\pi^k}(s_h^k, a_h^k) = V_{h,k}^{\pi^k}(s_h^k)$ by the definition. Thus,

$$\delta_h^k \leq \delta_{h+1}^k + \zeta_{h+1}^k + 2\beta \left\| \boldsymbol{\phi}_h^k \right\|_{(\Lambda_h^k)^{-1}} + B_{\boldsymbol{\theta},\mathcal{E}} \sqrt{d(k-\tau)} + B_{\boldsymbol{\mu},\mathcal{E}} H \sqrt{d(k-\tau)}.$$

$\square$

Now we are ready to prove Theorem 5, which is the regret upper bound within one epoch.

*Proof of Theorem 5.* We denote the dynamic regret within an epoch as Dyn-Reg$(\mathcal{E})$. We define $\delta_h^k = V_h^k(s_h^k) - V_{h,k}^{\pi^k}(s_h^k)$ and $\zeta_{h+1}^k = \mathbb{E}[\delta_{h+1}^k | s_h^k, a_h^k] - \delta_{h+1}^k$ as in Lemma 11. We derive the dynamic regret within a epoch $\mathcal{E}$ (the length of this epoch is $W$ which is equivalent to $\frac{W}{H}$ episodes) conditioned on the event $E$

defined in Lemma 1 which happens with probability at least $1 - p/2$.

$$
\begin{aligned}
&\text{Dyn-Reg}(\mathcal{E}) \\
&= \sum_{k \in \mathcal{E}} \left[ V_{1,k}^*(s_1^k) - V_{1,k}^{\pi^k}(s_1^k) \right] \\
&\leq \sum_{k \in \mathcal{E}} [V_1^k(s_1^k) + B_{\boldsymbol{\theta},\mathcal{E}} H \sqrt{d(k-\tau)} + B_{\boldsymbol{\mu},\mathcal{E}} H^2 \sqrt{d(k-\tau)} - V_{1,k}^{\pi^k}(s_1^k)] \\
&\leq \sum_{k \in \mathcal{E}} [\delta_1^k + B_{\boldsymbol{\theta},\mathcal{E}} H \sqrt{d(k-\tau)} + B_{\boldsymbol{\mu},\mathcal{E}} H^2 \sqrt{d(k-\tau)}] \\
&\leq \sum_{k \in \mathcal{E}} \sum_{h=1}^{H} \zeta_h^k + 2\beta \sum_{k \in \mathcal{E}} \sum_{h=1}^{H} \left\| \boldsymbol{\phi}_h^k \right\|_{(\Lambda_h^k)^{-1}} + 2 \sum_{k \in \mathcal{E}} B_{\boldsymbol{\theta},\mathcal{E}} H \sqrt{d(k-\tau)} + 2 \sum_{k \in \mathcal{E}} B_{\boldsymbol{\mu},\mathcal{E}} H^2 \sqrt{d(k-\tau)} \\
&\leq \sum_{k \in \mathcal{E}} \sum_{h=1}^{H} \zeta_h^k + 2\beta \sum_{k \in \mathcal{E}} \sum_{h=1}^{H} \left\| \boldsymbol{\phi}_h^k \right\|_{(\Lambda_h^k)^{-1}} + B_{\boldsymbol{\theta},\mathcal{E}} W \sqrt{2d(W/H+1)} + B_{\boldsymbol{\mu},\mathcal{E}} W \sqrt{2d(WH+H)} \quad (15)
\end{aligned}
$$

where the first inequality is due to Lemma 4, the third inequality is due to Lemma 11, and the last inequality is due to Jensen's inequality. Now we need to bound the first two terms in the right side. Note that $\{\zeta_h^k\}$ is a martingale difference sequence satisfying $|\zeta_h^k| \leq 2H$ for all $(k, h)$. By Azuma-Hoeffding inequality we have for any $t > 0$,

$$
\mathbb{P}\left( \sum_{k \in \mathcal{E}} \sum_{h=1}^{H} \zeta_h^k \geq t \right) \geq \exp(-t^2/(2WH^2)).
$$

Hence with probability at least $1 - p/2$, we have

$$
\sum_{k \in \mathcal{E}} \sum_{h=1}^{H} \zeta_h^k \leq 2H \sqrt{W \log(2dW/p)}. \quad (16)
$$

For the second term, note that by Lemma 14 for any $h \in [H]$, we have

$$
\sum_{k \in \mathcal{E}} (\boldsymbol{\phi}_h^k)^\top (\Lambda_h^k)^{-1} \boldsymbol{\phi}_h^k \leq 2 \log \left[ \frac{\det(\Lambda_h^{k+1})}{\det(\Lambda_h^1)} \right] \leq 2d \log \left( \frac{W}{H} + 1 \right).
$$

By Cauchy-Schwarz inequality, we have

$$
\begin{aligned}
\sum_{k \in \mathcal{E}} \sum_{h=1}^{H} \left\| \boldsymbol{\phi}_h^k \right\|_{(\Lambda_h^k)^{-1}} &\leq \sum_{h=1}^{H} \sqrt{W/H} \left[ \sum_{k \in \mathcal{E}} (\boldsymbol{\phi}_h^k)^\top (\Lambda_h^k)^{-1} \boldsymbol{\phi}_h^k \right]^{1/2} \\
&\leq H \sqrt{2d \frac{W}{H} \log \left( \frac{W}{H} + 1 \right)} \\
&\leq H \sqrt{2d \frac{W}{H} \log [2dW/p]}. \quad (17)
\end{aligned}
$$

Finally, combining Eq. 15–17, we have with probability at least $1 - p$,

$$
\begin{aligned}
\text{Dyn-Reg}(\mathcal{E}) &\leq 2H \sqrt{W \log(2dW/p)} + C_0 d H^2 \sqrt{\log(2dW)/p} \sqrt{2d \frac{W}{H} \log[2dW/p]} \\
&\quad + B_{\boldsymbol{\theta},\mathcal{E}} W \sqrt{2d(W/H+1)} + B_{\boldsymbol{\mu},\mathcal{E}} W \sqrt{2d(WH+H)} \\
&\lesssim \tilde{O}(\sqrt{d^3 H^3 W} + B_{\boldsymbol{\theta},\mathcal{E}} \sqrt{d/H} W^{3/2} + B_{\boldsymbol{\mu},\mathcal{E}} \sqrt{dH} W^{3/2}).
\end{aligned}
$$

$\square$

Now we can derive the regret bound for the whole time horizon by summing over all epochs and applying a union bound. We restate the regret upper bound and provide its detailed proof.

**Theorem 10.** *If we set $\beta = cdH\sqrt{\log(2dT/p)}$, the dynamic regret of* `LSVI-UCB-Restart` *algorithm is* $\tilde{O}(W^{-1/2}Td^{3/2}H^{3/2} + B_{\boldsymbol{\theta}}d^{1/2}H^{-1/2}W^{3/2} + B_{\boldsymbol{\mu}}d^{1/2}H^{1/2}W^{3/2})$, *with probability at least* $1 - p$.

*Proof.* In total there are $N = \lceil \frac{T}{W} \rceil$ epochs. For each epoch $\mathcal{E}_i$ if we set $\delta = \frac{p}{N}$, then it will incur regret $\tilde{O}(\sqrt{d^3H^3W} + B_{\boldsymbol{\theta},\mathcal{E}_i}\sqrt{d/H}W^{3/2} + B_{\boldsymbol{\mu},\mathcal{E}_i}\sqrt{dH}W^{3/2})$ with probability at least $1 - \frac{p}{N}$. By summing over all epochs and applying a union bound over them, we can obtain the regret upper bound for the whole time horizon. With probability at least $1 - p$,

$$\text{Dyn-Reg}(T) = \sum_{\mathcal{E}_i} \text{Dyn-Reg}(\mathcal{E}_i) \lesssim \sum_{\mathcal{E}_i} \tilde{O}(\sqrt{d^3H^3W} + B_{\boldsymbol{\theta},\mathcal{E}_i}\sqrt{d/H}W^{3/2} + B_{\boldsymbol{\mu},\mathcal{E}_i}\sqrt{dH}W^{3/2})$$

$$\lesssim \tilde{O}(d^{3/2}H^{3/2}TW^{-1/2} + B_{\boldsymbol{\theta}}d^{1/2}H^{-1/2}W^{3/2} + B_{\boldsymbol{\mu}}d^{1/2}H^{1/2}W^{3/2}).$$

$\square$

## C   Proofs in Section 5

In this section, we derive the regret bound for `Ada-LSVI-UCB-Restart` algorithm.

*Proof of Theorem 7.* Let $R_i(W, s_1^{(i-1)H})$ be the totol reward recieved in $i$-th block by running proposed `LSVI-UCB-Restart` with window size $W$ starting at state $s_1^{(i-1)H}$, we can first decompose the regret as follows:

$$\text{Dyn-Reg}(T) = \underbrace{\sum_{k=1}^{K} V_{1,k}^*(s_k^1) - \sum_{i=1}^{\lceil T/MH \rceil} R_i(W^\dagger, s_1^{(i-1)H})}_{\text{\textcircled{1}}} + \underbrace{\sum_{i=1}^{\lceil T/MH \rceil} (R_i(W^\dagger, s_1^{(i-1)H}) - R_i(W_i, s_1^{(i-1)H}))}_{\text{\textcircled{2}}},$$

where term \textcircled{1} is the regret incurred by always selecting the best epoch size for restart in the feasible set $J_W$, and term \textcircled{2} is the regret incurred by adaptively tuning epoch size by `EXP3-P`. We denote the optimal epoch size in this case as $W^* = \lceil (B_{\boldsymbol{\theta}} + B_{\boldsymbol{\mu}} + 1)^{-1/2}d^{1/2}H^{1/2}T^{1/2} \rceil H$. It is straightforward to verify that $1 \le W^* \le MH$, thus there exists a $W^\dagger \in J_W$ such that $W^\dagger \le W^* \le 2W^\dagger$, which well-approximates the optimal epoch size up to constant factors. Denote the total variation of $\boldsymbol{\theta}$ and $\boldsymbol{\mu}$ in block $i$ as $B_{\boldsymbol{\theta},i}$ and $B_{\boldsymbol{\mu},i}$ respectively. Now we can bound the regret. For the first term, we have

$$\text{\textcircled{1}} \lesssim \sum_{i=1}^{\lceil T/MH \rceil} \tilde{O}(d^{3/2}H^{3/2}MH(W^\dagger)^{-1/2} + B_{\boldsymbol{\theta},i}d^{1/2}H^{-1/2}(W^\dagger)^{3/2} + B_{\boldsymbol{\mu},i}d^{1/2}H^{1/2}(W^\dagger)^{3/2})$$

$$\lesssim \tilde{O}(d^{3/2}H^{3/2}T(W^\dagger)^{-1/2} + B_{\boldsymbol{\theta}}d^{1/2}H^{-1/2}(W^\dagger)^{3/2} + B_{\boldsymbol{\mu}}d^{1/2}H^{1/2}(W^\dagger)^{3/2})$$

$$\lesssim \tilde{O}(d^{3/2}H^{3/2}T(W^*)^{-1/2} + B_{\boldsymbol{\theta}}d^{1/2}H^{-1/2}(W^*)^{3/2} + B_{\boldsymbol{\mu}}d^{1/2}H^{1/2}(W^*)^{3/2})$$

$$\lesssim \tilde{O}((B_{\boldsymbol{\theta}} + B_{\boldsymbol{\mu}} + 1)^{1/4}d^{5/4}H^{5/4}T^{3/4}),$$

where the first inequality is due to Theorem 6, and the third inequality is due to $W^\dagger$ differs from $W^*$ up to constant factor. For the second term, we can directly apply the regret bound of `EXP3-P` algorithm (Bubeck & Cesa-Bianchi, 2012). In this case there are $\Delta = \ln M + 1$ arms, number of equivalent time steps is $\lceil \frac{T}{MH} \rceil$, and loss per equivalent time step is bounded within $[0, MH]$. Thus we have

$$\text{\textcircled{2}} \lesssim \tilde{O}(MH\sqrt{\Delta T/MH}) \le \tilde{O}(d^{1/4}H^{3/4}T^{3/4}).$$

Combining the bound of \textcircled{1} and \textcircled{2} yields the regret bound of `Ada-LSVI-UCB-Restart`,

$$\text{Dyn-Reg}(T) \lesssim \tilde{O}((B_{\boldsymbol{\theta}} + B_{\boldsymbol{\mu}} + 1)^{1/4}d^{5/4}H^{5/4}T^{3/4}).$$

$\square$

# D   Auxiliary Lemmas

In this section, we present some useful auxiliary lemmas.

**Lemma 12.** *For any fixed policy $\pi$, let $\{w_{h,k}^\pi\}_{h\in[H],k\in[K]}$ be the corresponding weights such that $Q_{h,k}^\pi(s,a) = \langle \phi(s,a), w_{h,k}^\pi \rangle$ for all $(s,a,h,k) \in \mathcal{S} \times \mathcal{A} \times [H] \times [K]$. Then we have*

$$\forall (k,h) \in [K] \times [H], \quad \left\| w_{h,k}^\pi \right\| \leq 2H\sqrt{d}.$$

*Proof.* By the Bellman equation, we know that for any $(h,k) \in [H] \times [K]$,

$$\begin{aligned}
Q_{h,k}^\pi(s,a) &= (r_h^k + \mathbb{P}_h^k V_{h+1,k}^\pi)(s,a) \\
&= \langle \boldsymbol{\theta}_{h,k} + \int V_{h+1,k}^\pi d\boldsymbol{\mu}_{h,k}(s'), \phi(s,a) \rangle \\
&= \langle w_{h,k}^\pi, \phi(s,a) \rangle,
\end{aligned}$$

where the second equality holds due to the linear MDP assumption. Under the normalization assumption in Definition 1, we have $\|\boldsymbol{\theta}_{h,k}\| \leq \sqrt{d}$, $V_{h+1,k}^\pi \leq H$ and $\|\boldsymbol{\mu}_{h,k}(s')\| \leq \sqrt{d}$. Thus,

$$w_{h,k}^\pi \leq \sqrt{d} + H\sqrt{d} \leq 2H\sqrt{d}.$$

$\square$

**Lemma 13.** *Let $\Lambda_t = I + \sum_{i=1}^t \boldsymbol{\phi}_i^\top \boldsymbol{\phi}_i$, where $\boldsymbol{\phi}_t \in \mathbb{R}^d$, then*

$$\sum_{i=1}^t \boldsymbol{\phi}_i^\top (\Lambda_t)^{-1} \boldsymbol{\phi}_i \leq d.$$

*Proof.* We have $\sum_{i=1}^t \boldsymbol{\phi}_i^\top (\Lambda_t)^{-1} \boldsymbol{\phi}_i = \sum_{i=1}^t \text{Tr}(\boldsymbol{\phi}_i^\top (\Lambda_t)^{-1} \boldsymbol{\phi}_i) = \text{Tr}((\Lambda_t)^{-1} \sum_{i=1}^t \boldsymbol{\phi}_i \boldsymbol{\phi}_i^\top)$. After apply eigenvalue decomposition, we have $\sum_{i=1}^t \boldsymbol{\phi}_i \boldsymbol{\phi}_i^\top = \mathbf{U}\text{diag}(\lambda_1, \ldots, \lambda_d)$ and $\Lambda_t = \mathbf{U}\text{diag}(\lambda_1 + 1, \ldots, \lambda_d + 1)$. Thus $\sum_{i=1}^t \boldsymbol{\phi}_i^\top (\Lambda_t)^{-1} \boldsymbol{\phi}_i = \sum_{i=1}^d \frac{\lambda_i}{\lambda_i} \leq d$.

$\square$

**Lemma 14.** *(Abbasi-Yadkori et al., 2011) Let $\{\boldsymbol{\phi}_t\}_{t\geq 0}$ be a bounded sequence in $\mathbb{R}^d$ satisfying $\sup_{t\geq 0} \|\boldsymbol{\phi}_t\| \leq 1$. Let $\Lambda_0 \in \mathbb{R}^{d \times d}$ be a positive definite matrix. For any $t \geq 0$, we define $\Lambda_t = \Lambda_0 + \sum_{j=1}^t \boldsymbol{\phi}_j^\top \boldsymbol{\phi}_j$. Then if the smallest eigenvalue of $\Lambda_0$ satisfies $\lambda_{\min}(\Lambda_0) \geq 1$, we have*

$$\log\left[\frac{\det(\Lambda_t)}{\det(\Lambda_0)}\right] \leq \sum_{j=1}^t \boldsymbol{\phi}_j^\top \Lambda_{j-1}^{-1} \boldsymbol{\phi}_j \leq 2\log\left[\frac{\det(\Lambda_t)}{\det(\Lambda_0)}\right].$$

# E   Details of the Experiments

## E.1   Synthetic Linear MDP Construction

The MDP has $S = 15$ states, $A = 7$ actions, $H = 10$, $d = 10$, and $T = 20000$, and 5 special chains. The states are denoted $s_1, \ldots, s_S$, and the actions are denoted $a_1, \ldots, a_A$. We first construct the known feature $\phi$. Intuitively, $\phi$ represents the transition from $\mathcal{S} \times \mathcal{A}$ space to $d$-dim space. We let the special chains have the correct transition to the $d$-dim space while other parts have random transition. This special transition will later be connected with $\boldsymbol{\mu}_{h,k}$, (i.e., the transition from $d$-dim space to $\mathcal{S}$ space) to form the chain. A special property of the construction is at any episode $k$, the transition function only has one connected chain and such chain is similar to combination lock. The agent must find this unique chain to achieve good behavior.

We let feature $\phi$ have "one-hot" form (each $(s,a)$ deterministically transits to a latent state in $d$-dim space), and satisfy the following:

1. For special chain $i = 1, \ldots, 5$, we have $\boldsymbol{\phi}(s_i, a_i)[i] = 1$ and $\boldsymbol{\phi}(s_i, a_i)[n] = 0, n \neq i$. For $i = 1, \ldots, 5$ and $j \in [A], j \neq i$, we have $\boldsymbol{\phi}(s_i, a_j)[n] = 1$ and $\boldsymbol{\phi}(s_i, a_j)[l] = 0, l \in [S], l \neq n$, where $k$ is uniformly drawn from $1, \ldots, i-1, i+1, \ldots, d$.

2. For normal chain $i = 6, \ldots, S$, and $j \in [A]$ we have $\boldsymbol{\phi}(s_i, a_j)[n] = 1$, and $\boldsymbol{\phi}(s_i, a_j)[l] = 0, l \in [S], l \neq n$, where $n$ is uniformly drawn from $1, \ldots, d$.

Now consider designing $\boldsymbol{\mu}_{h,k}$. As mentioned in the main text, we have a set of 5 different MDPs, and they have unique but different good chains. The *abruptly-changing* environment abruptly switches the good chain (or linear MDP) periodically every 100 episodes, whereas the *gradually-changing* environment switches the good chain (or linear MDP) continuously from one to another within every 100 episodes. In the following construction, we only design those 5 different MDPs for the *abruptly-changing* environment since we only need to take convex combination of different MDPs in the *gradually-changing* case. For each of those 5 linear MDPs, we only let one special chain be connected, and it becomes the good chain. Other special chains are broken in the $\boldsymbol{\mu}_{h,k}$ part (i.e., the part transits from $d$-dim space to $\mathcal{S}$ space). When the good chain is $g \in [5]$ in episode $k$, we let $\boldsymbol{\mu}_{h,k}$ satisfy the following:

1. For good chain $g$, $\forall h \in [H]$, we have $\boldsymbol{\mu}_{h,k}(s_g)[g] = 0.99$, $\boldsymbol{\mu}_{h,k}(s_{g+1})[g] = 0.01$, and $\boldsymbol{\mu}_{h,k}(s_n)[g] = 0, n \neq g, g+1$.

2. For other special but not good chain $\breve{g} \in [5], \breve{g} \neq g$, $\forall h \in [H]$, we have $\boldsymbol{\mu}_{h,k}(s_{\breve{g}})[\breve{g}] = 0.01$, $\boldsymbol{\mu}_{h,k}(s_{\breve{g}+1})[\breve{g}] = 0.99$, and $\boldsymbol{\mu}_{h,k}(s_n)[\breve{g}] = 0, n \in [S], n \neq \breve{g}, \breve{g}+1$.

3. For all normal chain $i = 6, \ldots, d$, $\forall h \in [H]$, we randomly sample two states $n_1$ and $n_2$, and let $\boldsymbol{\mu}_{h,k}(s_{n_1})[i] = 0.8$, $\boldsymbol{\mu}_{h,k}(s_{n_2})[i] = 0.2$, and $\boldsymbol{\mu}_{h,k}(s_n)[i] = 0, n \in [S], n \neq n_1, n_2$.

Finally we construct $\boldsymbol{\theta}_{h,k}$, which is related to the reward function. To ensure $g \in [5]$ is a good chain, we place a huge reward at the end of the chain but 0 reward for the rest of the chain. In addition, we put small intermediate rewards on sub-optimal actions. Specifically, when $g$ is the good chain at episode $k$, $\boldsymbol{\theta}_{h,k}$ is as follows:

1. For good chain $g$, we have $\boldsymbol{\theta}_{h,k}[g] = 0, \forall h \in [H-1]$, and $\boldsymbol{\theta}_{H,k}[g] = 1$.

2. For all other chains $i \in [d], i \neq g$, $\forall h \in [H]$, we let $\boldsymbol{\theta}_{h,k}[i]$ uniformly sample from $[0.005, 0.008]$.

In our construction, it is straightforward to verify that we have a valid transition function, and the transition function and reward function together satisfy the combination lock type construction. Notice that sometimes we refer to the normal chain in the $\mathcal{S}$ space and sometimes in the $d$-dim space. The reason is that a special chain must be connected in both parts ($\mathcal{S} \times \mathcal{A}$ space to $d$-dim space and $d$-dim space to $\mathcal{S}$ space), so we can break any part to make it a normal chain.

## E.2 Hardware Details

All experiments are performed on a Macbook Pro with 8 cores, 16 GB of RAM.

