# OpenReview forum: "Nonstationary Reinforcement Learning with Linear Function Approximation"
_TMLR — Accepted by TMLR_

### Review · Reviewer_8NKu · 2022-08-05

**Summary Of Contributions:**

This work studies regret minimization in linear (low-rank) MDPs when the environment is non-stationary. The primary contribution is several upper bounds in this setting—first in the case when the amount of non-stationarity is known, and then in the case when it is unknown, both of which scale as $O(B^{1/4} d^{5/4} H^{5/4} T^{3/4})$, for $B$ a measure of how much the environment changes. In addition to the upper bounds, a lower bound scaling as $\Omega(B^{1/3} d^{2/3} H^{1/3} T^{2/3})$ is shown. Finally, experiments are providing illustrating the proposed algorithm on a simple example.


**Requested Changes:**

- Is it possible to obtain a regret bound in terms of the total number of environment switches (e.g. the $L$ parameter of (Wei & Lou, 2021))? Even if it is not, I think this should be discussed, and the technical challenges that prevent the proposed approaches from obtaining this guarantee should be described.
- The experiments do not compare against the algorithms of (Touati & Vincent, 2020) and (Wei & Lou, 2021). I believe it is important that comparisons against these algorithms are presented.
- The MDP used in the experiments is fairly simple and almost a tabular MDP—the feature vectors are all one-hot encodings, though $d < SA$, so the MDP does still have a non-trivial low-dimensional structure—and the state and action spaces are relatively small (both under 20). It could be interesting to look at more complicated linear MDPs with larger state spaces (for example with feature vectors living in the simplex). I do not think this is a critical change but it would make the experiments section more compelling.
- The works [1,2] (see blow) both present lower bounds of $\Omega(d\sqrt{K})$ in the linear MDP setting and should be cited in relation to the lower bound given here ([2] just appeared at ICML 2022 and is concurrent).
- The definition of environment change, $B_\mu$, seems perhaps overly restrictive. Given the nature of the linear MDP setting, where there are potentially an infinite number of states and the probability of reaching any one of them could be 0, one could easily have that $B_\mu$ is arbitrarily large, but the actual MDP the user interacts with is identical except for on a measure 0 set. In the worst case this is tight, and this is also the definition of environment change considered by (Touati & Vincent, 2020) and (Wei & Lou, 2021), but it seems possible to do better. For example, perhaps one could measure change in a way that takes into account how easily certain states can be reached. This is not an essential change but would strengthen the work.


Minor typos:
- At the top of page 13, “thw” should be “the”.
- In section 7, I believe the sentence “We derived a minimax regret lower bound is for nonstationary…” should read “We derived a minimax regret lower bound for nonstationary…”.


[1] Zanette, Andrea, et al. "Learning near optimal policies with low inherent bellman error." International Conference on Machine Learning. PMLR, 2020.

[2] Hu, Pihe, Yu Chen, and Longbo Huang. "Nearly Minimax Optimal Reinforcement Learning with Linear Function Approximation." International Conference on Machine Learning. PMLR, 2022.

**Strengths And Weaknesses:**

Strengths:
- The paper is easy to follow and the results are clearly laid out. The algorithms are relatively intuitive. I found the exposition intuitive and compelling, with the authors gradually building up the sophistication of the algorithm as assumptions are relaxed.
- From what I can tell the results appear technically sound—I read through the proofs and all seem correct.
- To the best of my knowledge, the lower bound on time-variations in the linear MDP setting is novel.
- Experimental results demonstrate that the proposed algorithms improve on baselines which do not take into account the time variation.
- In the case when “local variation” is known, it is shown that the proposed algorithm achieves the optimal $B$ and $T$ dependence of $O(B^{1/3} d^{4/3} H^{4/3} T^{2/3})$. Here known “local variation” means that the learner knows not just the total change over all $T$ steps, but also the change in each individual epoch of the algorithm. While knowing this local variation is a relatively strong and unrealistic assumption, it is interesting that it is possible to obtain the optimal rate with this knowledge.

Weaknesses:
- The regret bound presented in this work scales as a suboptimal $O(B^{1/4} T^{3/4})$, off the optimal rate of $O(B^{1/3} T^{2/3})$ (ignoring $d$ and $H$ dependence).
- There is significant overlap between this work and existing work. A paper from 2020, (Touati & Vincent, 2020), achieves an identical regret bound in an identical setting (though it lacks a lower bound and the result in the “local variation” setting). A paper from early 2021, (Wei & Lou, 2021), achieves a tighter regret bound (achieving regret of $O(B^{1/3} d^{4/3} H^{4/3} T^{2/3})$, which has optimal $B$ and $T$ dependence, as well as hitting a regret bound in terms of the total number of switches, $L$) and applies in much more general settings. Both of these works are discussed in related work.
- The techniques are relatively standard. In particular, the basic algorithm presented, and analysis, are very similar to the LSVI-UCB algorithm of (Jin et al., 2020), and then simply apply the Bandit-over-Bandit strategy of (Cheung et al., 2019) to handle non-stationarity. As such, I am not certain the algorithmic techniques are of significant interest to the community.

---

> ### Author Response · Authors · 2022-08-27
> **Response to Reviewer 8NKu**
>
> We thank Reviewer 8NKu for recognizing our presentation is clear, the result to be technically solid, and appreciating our novel lower bound on time-variations in the linear MDP setting, experimental results and "local variation" results. Please find our response below.
>
> -----
> 1\. Q: "The regret bound presented in this work is suboptimal with respect to total variation $B$ and total number of time steps $T$."
>
> A: Please refer to bullet points 2 and 3 in the general response.
>
> -----
> 2\. Q: "There is significant overlap between this work and existing work. A paper from 2020, (Touati & Vincent, 2020), achieves an identical regret bound in an identical setting (though it lacks a lower bound and the result in the “local variation” setting). A paper from early 2021, (Wei & Lou, 2021), achieves a tighter regret bound (achieving regret of $\tilde{O}(d^{4/3}H^2B^{1/3}T^{2/3})$, which has optimal $B$ and $T$ dependence, as well as hitting a regret bound in terms of the total number of switches, ) and applies in much more general settings. Both of these works are discussed in related work."
>
> A: Please refer to bullet points 2 and 3 in general response.
>
> -----
> 3\. Q: "The techniques are relatively standard. In particular, the basic algorithm presented, and analysis, are very similar to the LSVI-UCB algorithm of (Jin et al., 2020), and then simply apply the Bandit-over-Bandit strategy of (Cheung et al., 2019) to handle non-stationarity. As such, I am not certain the algorithmic techniques are of significant interest to the community."
>
> A: Please refer to bullet points 2 and 3 in the general response. Although our algorithmic ideas are adapted from existing works, the analysis of the dynamic regret upper bound is not a straightforward extensions of existing works. Together with the lower bound results and the numerical results, we believe our results are of interest to the community.
>
>
> -----
> 4\. Q: "Is it possible to obtain a regret bound in terms of the total number of environment switches (e.g. the $L$ parameter of (Wei & Lou, 2021))? Even if it is not, I think this should be discussed, and the technical challenges that prevent the proposed approaches from obtaining this guarantee should be described."
>
> A: Our algorithm can still achieve a regret bound in terms of the total number of environment changes, but with a sub-optimal rate $\tilde{O}(L^{1/3}T^{2/3})$. The reason is periodic restart strategy is not suitable to handle abrupt changes since the passive nature indicates that we cannot guarantee to detect the abrupt environment change within a reasonably short delay. Wei & Luo overcome this issue by running two tests on top of multiple base instances to detect the environmental change. Similar ideas have also been used in the piecewise-stationary bandit literature, where a change detection subroutine is run to detect the environmental change, so that the regret incurred by the environmental drift can be controlled.
>
> We have added the related discussions in Remark 4 of Section 4.2.1 in the revised version.
>
> -----
> 5\. Q: "The experiments do not compare against the algorithms of (Touati & Vincent, 2020) and (Wei & Luo, 2021). I believe it is important that comparisons against these algorithms are presented."
>
> A: We have implemented the algorithms of Touati & Vincent (2020) and Wei & Luo (2021) as baselines in the revision. Please check Section 6 in the revised version for details. Our algorithm achieves comparable performance compared to these two algorithms, but it is much more computationally efficient. The reason is our algorithm only requires the most recent trajectories to estimate the $Q$-function, whereas the others requires the entire history. In addition, the algorithm in Wei & Luo (2021) requires maintaining multiple base instances at different scales, which further increases the computational burden.

---

> > ### Author Response · Authors · 2022-08-27
> > **Response to Reviewer 8NKu Continued**
> >
> > -----
> > 6\. Q: "The MDP used in the experiments is fairly simple and almost a tabular MDP—the feature vectors are all one-hot encodings, though $d<SA$, so the MDP does still have a non-trivial low-dimensional structure—and the state and action spaces are relatively small (both under 20). It could be interesting to look at more complicated linear MDPs with larger state spaces (for example with feature vectors living in the simplex). I do not think this is a critical change but it would make the experiments section more compelling."
> >
> > A: Thanks for the comment. Firstly, we want to highlight that we are the first to report experiments in the nonstationary linear MDP setting. The two concurrent works by Touati & Vincent and Wei & Luo do not have empirical investigation. We choose one-hot encodings to make the (exploration) problem in nonstationary linear MDP harder since it shares a similar difficulty of the combinatation lock. In the experiment, our right components in the reward function and transition function decomposition are not one-hot and already lie in the simplex as suggested by the reviewer.
> >
> > The main obstacle for large-scale experiment is the high inefficiency of the baseline algorithms that we compare (e.g., LSVI-UCB, OPT-WLSVI, Epsilon-Greedy already take around 2 hours in such a small scale environment, and Master takes even longer, which is roughly 12 hours). Our experiment mostly serves as a proof of concept and we believe that one can easily see from the figures that our algorithms achieve both smaller regret and significantly less runtime. Since our paper is already fairly long, we leave further empirical studies as future work.
> >
> > -----
> > 7\.
> > Q: "The works [1,2] (see blow) both present lower bounds of $\Omega(d \sqrt K)$ in the linear MDP setting and should be cited in relation to the lower bound given here ([2] just appeared at ICML 2022 and is concurrent)."
> >
> >
> > A: Thanks for the references! We have included in Section 3 when proving the new lower bound for heterogeneous setting.
> >
> >
> >
> > -----
> > 8\. Q: "The definition of environment change, $B_{\mu}$, seems perhaps overly restrictive. Given the nature of the linear MDP setting, where there are potentially an infinite number of states and the probability of reaching any one of them could be 0, one could easily have that $B_{\mu}$ is arbitrarily large, but the actual MDP the user interacts with is identical except for on a measure 0 set. In the worst case this is tight, and this is also the definition of environment change considered by (Touati & Vincent, 2020) and (Wei & Lou, 2021), but it seems possible to do better. For example, perhaps one could measure change in a way that takes into account how easily certain states can be reached. This is not an essential change but would strengthen the work."
> >
> > A: Thanks for the great comment! As indicated by our lower bound, the current definition of the environment change is inevitable in the worst case, so the definition of $B_\mu$ works for all linear MDPs. It is possible to define a new environment change measure by imposing some additional constraints on the underlying linear MDPs. The environment change measure comes into play when we derive the error of the action-value function estimate, where we bound the error term contributed by the environment change using the worst case. The current analysis still goes through if the unreachable states are the same across all episodes, which in some sense is limited. In addition, such reaching probability in general would also change in the nonstationary environment; therefore, it is hard to define and capture that. We leave defining a new environment change measure as an interesting future direction.
> >
> > We have added Remark 1 to discuss this issue at the end of Section 2.1 in the revised version.
> >
> > -----
> > 9\. Q: "Minor typos"
> >
> > A: Thanks for the detailed reading! We have fixed them in the revision.
> >
> > -----
> > We hope our response addresses your concerns. Please let us know if you have further questions, and hope you can adjust your score and the formal decision recommendation accordingly.

---

### Review · Reviewer_4EKk · 2022-08-16

**Summary Of Contributions:**

This paper studies non-stationary linear MDPs with the following contributions:
1. A minimax lower bound, which also implies a lower bound for standard linear MDP.
2. A new algorithm with upper bounds in the known variation and unknown variation setting. For the known variation setting, the bound is tight for T and B.
3. A parameter-free algorithm for the unknown variation setting with the same theoretical guarantee.
4. Numerical experiments to demonstrate the effectiveness of the proposed algorithms.

**Requested Changes:**

I suggest the following two changes:
1. Improve the bound to T^{2/3}, so as to match the state-of-the-art result.
2. Use the recent work on minimax linear MDP to improve the dependency on H and d: https://arxiv.org/abs/2206.11489

**Strengths And Weaknesses:**

Strengths:
1. Both upper bounds and lower bounds are presented.
2. The parameter-free algorithm is nice.
2. Numerical experiments are conducted, which is nice to have for a theoretical RL paper.

Weaknesses:
1. The upper and lower bounds do not match. Furthermore, the upper bound is worse than a prior work by Wei & Luo.
2. The techniques used are quite standard in the literature.

---

> ### Author Response · Authors · 2022-08-27
> **Response to Reviewer 4EKk**
>
> We would like to thank Reviewer 4EKk for appreciating our presentation about upper bounds, lower bounds, the parameter-free algorithm, and numerical experiments. Please find our response below.
>
> -----
> 1\. Q: "The upper and lower bounds do not match. Furthermore, the upper bound is worse than a prior work by Wei & Luo."
>
> A: Please refer to bullet points 1, 2, and 3 in the general response.
>
>
> 2\. "The techniques used are quite standard in the literature."
>
> A: Please refer to bullet points 2 and 3 in the general response.
>
> -----
> 2\. Q: "Improve the bound to T^{2/3}, so as to match the state-of-the-art result."
>
>
> A: Please refer to the bullet points 2 and 3 in the general response. In addition, we would like to highlight that our algorithm has advantages over related works in terms of computational efficiency. The work by Touati & Vincent (2020) requires the data in the entire history to compute the linear Q function estimator, and the work by Wei & Luo (2021) requires maintaining multiple base algorithm instances with different length to handle the non-stationarity. In contrast, the periodic restart strategy only requires the most recent trajectories to compute the Q function estimate.
>
> -----
> 3\. A: "Use the recent work on minimax linear MDP to improve the dependency on H and d: https://arxiv.org/abs/2206.11489"
>
> A: Thanks for pointing out the recent work on tighter minimax regret bound for stationary linear MDPs, we have added another dynamic regret lower bound based on this work. Please refer to bullet point 1 in the general response. We would also like to highlight that we are the first to provide the minimax dynamic regret lower bound for nonstationary linear MDPs, as noted by Reviewer 8NKu.
>
>
>
> -----
> We hope our response addresses your concerns. Please let us know if you have further questions, and hope you can adjust your score and the formal decision recommendation accordingly.

---

### Review · Reviewer_Kk4U · 2022-08-18

**Summary Of Contributions:**

This paper considers the problem of reinforcement learning in the linear MDP setting where the transition probabilities and rewards are allowed to change over time. In this setting, one aims to prove dynamic regret guarantees that bound the cumulative sub-optimality gaps between the optimal policy and the learned policy (on the round-varying MDP). In settings like this, one expects to have dependence on the total amount of variation in the problem (amid the changes in the transitions and reward). The authors propose a lower bound $O((dT)^{2/3} B^{1/3})$ where $B$ is a bound on the total variation across all the rounds.

An algorithm based on a restarting version of LSVI-UCB is proposed. The paper first starts with the case where “local variations” or the variation along any particular window/epoch is known. This allows one to choose windows to run LSVI-UCB and tune the optimism and the size of the window based on the budget. The rate nearly matches the lower bound

Another algorithm is given where the local variations are not known, but this requires inflating the optimism and leads to a slightly worse bound.

A final adaptive algorithm is proposed, which is designed to automatically handle the case where the variations are known. This is done via a meta algorithm that break the full $K$ rounds down into length $M$ chunks. Then, for each chunk, a window size $W$ is selected to run the LSVI-UCB restart algorithm over the length $M$ chunk. The size is selected based on an Exp3 meta algorithm.

Toy experiments are provided showing the effectiveness of the algorithms vs stationary methods.


**Broader Impact Concerns:**

There are no immediate concerns.

**Requested Changes:**

This paper is already in good shape technically for the most part. My suggested changes and questions are primarily concerning clarity and are here for  the purpose of strengthening the paper.



I don’t know if the term “local variation” is common in some of the immediate prior works, but I am personally not familiar with it and found it confusing when it was mentioned several times in the introduction without a proper or at least intuitive definition. I recommend clarifying that.


The paper mentions two concurrent works: Touati & Vincent and Wei & Luo. Evidently Wei & Luo, prove a better bound, but the algorithmic differences and computational disadvantages of their algorithm is described in the current paper. However, the method of Touati & Vincent, which gets the same rate as the current paper, is not described. I think it would be helpful to describe the main idea of their result and differentiate it from the current work. It would also be helpful to explicitly state the bounds achieved in these papers.

I’m not sure the result of Theorem 1 _that_ significant, as claimed. From the $d\sqrt{T}$ already comes up from existing linear bandit literature and this seems to be reframing this an MDP so that one can scale it up by H if I’m not mistaken. The proof of the theorem may be technically interesting, but it’s also a bit of a distraction from the non-stationary setting. I think the paper would be in good shape even without it.

Can the results of Zhong et al be applied to this setting or the other way around? Or do these two problem classes not contain each other?


I don’t doubt that Lemma 1 is true but I think the proof is not sufficient. Even if it is a minor modification of Jin et al, I think it’s worth showing explicitly how that happens. I suspect that one can apply their Lemma D.4 directly to do this.

For Lemma 1, it would be good to state the specific lemma it modifies from Jin et al in the main paper (I see that it’s listed in the appendix already).
In Definition 1 should the norm on $\mu$ be for all $s’$?

I may have missed it but it doesn’t appear the budget $B$ is defined before it’s used in Theorem 2.

In 4.1 I would recommend including a note that much of this is following the algorithmic principles of UCB or LSVI-UCB. The key difference seems to be setting the confidence parameter and the use of epochs.

I’m skeptical of Remark 3. I am not aware of a particular instance where the $d^{3/2}$ for LSVI-UCB is tight so it is conceivable that an improvement could be possible. Unless the authors can point out a reference to a particular case, it might be better to rephrase this as saying that the $d^{3/2}$ is unlikely to improve unless there is an improvement to LSVI-UCB.

In the first figure of the experiments, if you run LSVI-UCB-restart-unknown for more time steps, would you expect it to eventually outperform vanilla LSVI-UCB?

**High level questions that I’d like to see addressed in the revision**

I have several additional questions that occurred to me while reviewing this.

The use of the master algorithm Exp3 to handle selection of the window size is very reminiscent of “corralling” or model selection approaches to online RL [1, 2, 3, 4]. However, each of these requires some sort of “trick” (e.g. stochasticity or some significant forced exploration) and the rates are usually worse. I am wondering: why, fundamentally, is this not an issue in this paper? I hypothesize the reason is that you do not have a “starvation” problem identified in [1] since each chunk is an independent restarted run. However, I believe the authors may be in a better position to provide insights about this relationship.

[1] Agarwal, A., Luo, H., Neyshabur, B. and Schapire, R.E., 2017, June. Corralling a band of bandit algorithms. In Conference on Learning Theory (pp. 12-38). PMLR.

[2] Pacchiano, A., Phan, M., Abbasi Yadkori, Y., Rao, A., Zimmert, J., Lattimore, T. and Szepesvari, C., 2020. Model selection in contextual stochastic bandit problems. Advances in Neural Information Processing Systems, 33, pp.10328-10337.

[3] Lee, J., Pacchiano, A., Muthukumar, V., Kong, W. and Brunskill, E., 2021, March. Online model selection for reinforcement learning with function approximation. In International Conference on Artificial Intelligence and Statistics (pp. 3340-3348). PMLR.

[4] Abbasi-Yadkori, Y., Pacchiano, A. and Phan, M., 2020. Regret balancing for bandit and rl model selection. arXiv preprint arXiv:2006.05491.



Additionally, I wonder if it is possible to simply treat the variation as misspecification and simply apply Theorem 3.2 from Jin et al setting the misspecification parameter to the value of the local variation. Probably you would still have to use epochs, but is this significantly different from the proposed algorithm? If such a modification would work, I am curious if it would be possible to then apply results similar to [5] to then adapt the misspecification. It would be nice to include some discussion of the former in the paper, but the latter is just my own curiosity.

[5] Foster, D.J., Gentile, C., Mohri, M. and Zimmert, J., 2020. Adapting to misspecification in contextual bandits. Advances in Neural Information Processing Systems, 33, pp.11478-11489.



**Strengths And Weaknesses:**

**Strengths**

Overall I think this is a good paper.

To my knowledge, the proposed algorithms and their analyses are new. They are simple and appear to provide fairly strong guarantees for the non-stationary problem. The lower bound also helps paint a comprehensive picture of the hardness of the problem, but it is not totally complete yet. While the algorithms are based on existing techniques (LSVI-UCB + epochs for non-stationarity) they appear to be used in a unique way to address this problem.

The discussion of related work appears to be thorough, but I cannot be 100% sure. There were a couple of points that were not clear to me mentioned in the next box.

The proofs generally follow the outline of proofs in Jin et al with some modifications to deal with the non-stationarity. Though the technical novelty of the modifications is small, the simplicity is desirable. I have read all the proofs but not to the degree of checking constants. I think it is sound.

The experiments are pretty informative, but they are toy experiments designed to fit the problem setting.

**Weaknesses**

It is clear that there are several things that have gone unaddressed. For example, the adaptive algorithms have worse regret rates than the stated lower bound. So something is not tight, but it is not clear whether it is the lower bound or the algorithm. Nonetheless, this paper has already provided a number of non-trivial contributions, so it makes sense that this could be addressed in future work instead. I don’t think this is a major drawback of the paper and it was discussed by the authors in the conclusion.

While the paper is generally well written and easy to understand, there are a few standout places where the clarity could definitely be improved. I suspect that these can be fixed easily, assuming that the authors agree. I have listed these in the next box.

---

> ### Author Response · Authors · 2022-08-27
> **Response to Reviewer Kk4U**
>
> We thank Reviewer Kk4U for appreciating our work, providing valuable and constructive feedback, and especially for the advice on how to further strengthen the paper. Please find our response below.
>
>
> -----
> 1\. Q: Clarify the definition of local variation
>
> A: Thanks for the comment. We now formally introduce the definition of local variation in the introduction section and add a pointer to equation (2) in Section 4.2. Please refer to the revised manuscript for the details.
>
> -----
> 2\. Q: Describe the main result and idea of Touati & Vincent and differentiate it from the current work.
>
> A: We have added more discussion on the work by Touati & Vincent in the revised version in the introduction section. The key difference in the algorithm design is that their algorithm uses weighted least squares value iteration to handle the nonstationarity of the environment. In our setting, we use periodic restart to handle the nonstationarity.
>
> -----
> 3\. Q: Signification of Theorem 1 (maybe remove it)
>
> A: We have moved Theorem 1 from the main paper to the appendix. We have also added a new dynamic regret lower bound for the inhomogeneous setting in the revised version, where the transition function $P_h^k$ (as introduced in Section~\ref{sec:intro}) can be different for different $h$, based on the hard instance construction in [1, 2]. Note that the previous lower bound we provided is derived for the homogeneous setting (a special case of the inhomogeneous setting), where the transition function $P_h^k$ will be the same within an episode, i.e., for any $k$, $P_h^k\equiv P^k$ for any $h=\{1,\ldots,H\}$. Please also refer to the first bullet point in our general response.
>
> -----
> 4\. Q: Can the results of Zhong et al be applied to this setting or the other way around? Or do these two problem classes not contain each other?
>
> A: Zhong et al work on a different set of MDPs, namely linear kernel MDPs. Their setting cannot be applied to ours since linear MDPs cannot imply linear kernel MDPs, and vice versa. Please check Section 3 in [3] for a detailed discussion. The key difference is that for linear kernel MDPs, the feature map for transition functions is related to both the current state, action, and the next state. For linear MDPs, the feature map for transition function only depends on the current state and action.
>
> -----
> 5\. Q: Add the detailed proof for Lemma 1 and include the relevant lemmas in Jin et al.
>
> A: We have included the detailed proofs and relevant lemmas in Jin et al. in the revised appendix. In summary, we apply Lemma D.4 and Lemma D.6 in Jin et al. to prove our Lemma 1.
>
> -----
> 6\. Q: Norm on $\mu$ in Definition 1.
>
> A: Thanks for pointing out the typo. Yes, the norm should be defined over all states. We have corrected this in the revised version for Definition 1 and now use the same notation as that in Jin et al.
>
> -----
> 7\. Q: Definition of $B$
>
> A: We have defined the total variation $B$ right after Definition 1, as the summation of $B_\mu$ and $B_\theta$.
>
> -----
> 8\. Q: Including notes to discuss the relations between the proposed algorithm and UCB-based algorithm, also highlight the difference.
>
> A: We have included the notes at the end of Section 4.1 to discuss this matter.
>
> -----
> 9\. Q: Rephrase remark 3.
>
> A: Thanks for the suggestion, we have rephrased it accordingly. Please check the revised version (now it becomes remark 2).
>
> -----
> 10\. Q: Can LSVI-UCB-Restart-Unknown beat LSVI-UCB?
>
> Yes, LSVI-UCB-Resetart-Unknown can beat LSVI-UCB. We realized that in our previous implementation, we set the wrong epoch size $W$ to LSVI-UCB-Restart-Unknown. The previous value of $W$ is close to the total number of time steps $T$, so this is the reason why in the previous figure, the performance of LSVI-UCB-Restart-Unknown is close to LSVI-UCB in the first figure. Now we set the correct $W$ to LSVI-UCB-Restart-Unknown, and as a result the gap between these two algorithms is no longer trivial. Please check Figure in Section 6 in the revised version for details.

---

> > ### Author Response · Authors · 2022-08-27
> > **Response to Reviewer Kk4U Continued**
> >
> > -----
> > 11\. Q: Why there is no regret degradation when applying EXP3-P as the meta algorithm.
> >
> > A: Thanks for the great question! The fundamental reason is that we reduce the epoch size selection problem into an adversarial bandit problem with adaptive adversary. By properly optimizing the length of each block $M$, we can control the loss scale for epoch size selection. In addition, since we have discretized the candidate set of epoch size $J_W$ at a proper granularity, the best epoch size in $J_W$ can approximate the optimal epoch size up to constant factor. In other words, we have the freedom to choose the base algorithm, where each base algorithm is parameterized by one epoch size in $J_W$. However, many problem settings considered in the model selection literature do not have that much freedom to choose the base algorithm.  As a result, the additional regret incurred by master \texttt{EXP3-P} algorithm with respect to the best epoch size in $J_W$ can be controlled if we optimize the candidate set for epoch size $J_W$ and block length $M$ properly. Similar $\tilde{O}(B^{1/4}T^{3/4})$ regret overhead due to master algorithm also appears in nonstationary contextual bandit and tabular MDP literatures [1,2]. We have added Remark 6 in Section 5.2 to discuss why there is no regret degradation when applying EXP3-P as the meta algorithm.
> >
> >
> > -----
> > 12\. Q: Connection to misspecified linear MDPs.
> >
> > A: Thanks for the question and the reference! In early stages of our research, we had also thought about related approaches. By applying Cauchy-Schwarz inequality, one can show that our definition of nonstationary implies that misspecification in equation (4) of Jin et al. is also bounded (but not vice versa). However, there is a major challenge to use such a modification. The regret analysis in the misspecified linear MDP in Jin et al. is restricted to the static regret. In contrast, we consider the dynamic regret defined equation (1) in Section 2. More specifically, in the misspecified case, Jin et al. still assume the MDP is the same throughout the entire process and calculate the regret on such a fixed MDP. In our case, the MDP is always changing (nonstationary) and therefore the interaction with the environment and the regret calculation is on different MDPs.
> >
> > To handle the nonstationarity, we use different analysis. More specifically, we decompose the error in the regularized least squares estimation (our Lemma 2) and analyze the additional error due to environmental drift compared to the stationary setting. As a result, our Q function estimate is an optimistic upper bound of the optimal Q function (our lemma 3). In contrast, the Q function estimate for the misspecfied setting is an approximate optimistic upper bound up to some gap proportional to the misspecification error (Lemma C.6 in Jin et al.).
> >
> > We have also added related discussions in Remark 3 in section 4.2.1.
> >
> > -----
> > References
> >
> > [1] Wang Chi Cheung, David Simchi-Levi, and Ruihao Zhu. "Learning to optimize under non-stationarity." The 22nd International Conference on Artificial Intelligence and Statistics. PMLR, 2019.
> > [2] Weichao Mao, et al. "Near-optimal model-free reinforcement learning in non-stationary episodic mdps." International Conference on Machine Learning. PMLR, 2021.
> >
> > [3] Dongruo Zhou, Jiafan He, and Quanquan Gu. "Provably efficient reinforcement learning for discounted MDPs with feature mapping." International Conference on Machine Learning. PMLR, 2021.
> >
> > -----
> > We hope our response addresses your concerns. Please let us know if you have further questions, and hope you can adjust your score and the formal decision recommendation accordingly.

---

### Author Response · Authors · 2022-08-27
**General Response to the Reviewers**

We thank all three reviewers for their constructive and insightful feedback. Here we provide general responses to some common questions raised by different reviewers.

-----
1\. **Lower bound**

As suggested by reviewers 4EKk and 8NKu, we add a new minimax dynamic regret lower bound of order $\Omega(B^{1/3}d^{5/6}HT^{2/3})$ (Theorem 2 in the revised version) for the setting of inhomogeneous transition function, where the transition function $P_h^k$ (as introduced in Section 2 can be different for different $h$, based on the hard instance construction in [1, 2]. Note that the previous lower bound we provided is derived for the homogeneous setting (a special case of the inhomogeneous setting), where the transition function $P_h^k$ will be the same within an episode, i.e., for any $k$, $P_h^k\equiv P^k$ for any $h=\{1,\ldots,H\}$. For the details, please refer to Section 3 and Appendix A in the revised version. Note that the our proposed algorithms is designed for the inhomogeneous setting, since we maintain different Q function estimates for different steps.

-----
2\. **Discussion on the upper bound and novelty**

We are aware that Wei & Luo's work gives a tighter upper dynamic regret bound. However, the main goal of this paper is to provide a computationally-efficient algorithm for the setting of non-stationary linear MDPs. Compared to the work of Wei & Luo, our algorithm only requires maintaining one base LSVI-UCB instance, whereas they require $\log T$ base LSVI-UCB instances. Our algorithm can also achieve the $\tilde{O}(B^{1/3}T^{2/3})$ upper bound under the more restricted setting where we have knowledge of local variations. We leave further improvement to the regret of our algorithm while maintaining computational efficiency as future work.

Please note that although our algorithms are adapted from [3] and [4], we would like to emphasize the proof of the regret upper bound is not a straightforward extension of existing works. In particular, when deriving the upper bound of the least-squares estimation error (Lemma 2), we must carefully control the additional error terms incurred by the environment drift. Compared to [4], the additional error term incurred by environment drift is much more challenging to analyze since the error can propagate through the whole episode in an arbitrary manner. Contrarily for the bandit problem, one need not consider the error propagation problem due to unit planning horizon.

-----
3\. We would like to quote the policy of TMLR (https://www.jmlr.org/tmlr/reviewer-guide.html): "Crucially, it should not be used as a reason to reject work that isn't considered “significant” or “impactful” because it isn't achieving a new state-of-the-art on some benchmark. Nor should it form the basis for rejecting work on a method considered not “novel enough”, as novelty of the studied method is not a necessary criteria for acceptance. We explicitly avoid these terms (“significant”, “impactful”, “novel”), and focus instead on the notion of “interest”. If the authors make it clear that there is something to be learned by some researchers in their area from their work, then the criteria of interest is considered satisfied. TMLR instead relies on certifications (such as “Featured” and “Outstanding”) to provide annotations on submissions that pertain to (more speculative) assertions on significance or potential for impact."

Instead, the evaluation is based on "Are the claims made in the submission supported by accurate, convincing and clear evidence?" and "Would some individuals in TMLR's audience be interested in the findings of this paper?"

As acknowledged by different reviewers, we believe our paper is technically sound and the presentation is clear. We believe our work will be of interest to researchers working on decision making in non-stationary environments, and our contributions and novelty are at least modest and meet the acceptance standard of TMLR.

---

> ### Author Response · Authors · 2022-08-27
> **Rest of the General Response**
>
>
> -----
> 4\. We have prepared a revision. We summarize the changes we made in the revision (**Highlighted in red color in the revised version.** ) below:
>
>
> (a) We fixed the typos and improved the clarity of the paper based on reviewers' suggestions. Specifically, we add the following remarks to further strengthen the presentation: (1) Remark 1 to discuss the definition of nonstationarity measure $B_{\mu}$, (2) Remark 3 to discuss the connection to the misspecified setting in [3], (3) Remark 4 to discuss the dynamic regret bound for the case when there are $L$ abrupt switches, and (4) Remark 6 to discuss the connection between \texttt{Ada-LSVI-UCB-Restart} and model selection.
>
> (b) We added a new lower bound result. The new minimax dynamic regret lower bound is of order $\Omega(B^{1/3}d^{5/6}HT^{2/3})$ (Theorem 2 in the revised version) for the setting of inhomogeneous transition function, where the transition function $P_h^k$ can be different for different $h$, based on the hard instance construction in [1, 2]. Please refer to bullet point 1 for more details.
>
> (c) We provided additional numerical experiments to demonstrate our theoretical results, and we also implemented OPT-WLSVI [5] and MASTER [6] algorithms as baselines. Compared to OPT-LSVI and MASTER, our algorithm achieves comparable cumulative rewards but at a significantly smaller computational cost.
>
> (d) We added more discussions and remarks on the connection to existing works and definition on total variation, suggested by the reviewers.
>
> (e) We added more relevant references suggested by the reviewers.
>
> (f) Note that for minor typos, we may not mark them in the revised version.
>
> -----
> References
> [1] Dongruo Zhou, Quanquan Gu, and Csaba Szepesvari. "Nearly minimax optimal reinforcement learning for linear mixture Markov decision processes." Conference on Learning Theory. PMLR, 2021.
>
> [2] Pihe Hu, Yu Chen, and Longbo Huang. "Nearly minimax optimal reinforcement learning with linear function approximation." International Conference on Machine Learning. PMLR, 2022.
>
> [3] Chi Jin, Zhuoran Yang, Zhaoran Wang, and Michael I Jordan. "Provably efficient reinforcement learning with linear function approximation." Proc. Conf. Learning Theory (COLT), pages 2137–2143, July 2020.
>
> [4] Peng Zhao, Lijun Zhang, Yuan Jiang, and Zhi-Hua Zhou. "A simple approach for non-stationary linear bandits." Proc. 23rd Int. Conf. Artif. Intell. Stat. (AISTATS 2020), pages 746–755, August 2020.
>
> [5] Ahmed Touati and Pascal Vincent. "Efficient learning in non-stationary linear Markov decision processes." arXiv:2010.12870 (2020).
>
> [6] Chen-Yu Wei and Haipeng Luo. "Non-stationary reinforcement learning without prior knowledge: An optimal black-box approach."  Conference on Learning Theory. PMLR, 2021.

---

### Decision · Action_Editors · 2022-10-12

**Recommendation:** Accept as is

**Comment:**

The paper was deemed to be technically sound. The novelty of the contributions was contested by some reviewers, especially in light of some preprints that have been online since as early as 2020. Then again, a detailed look at the submission history of this paper reveals that this paper has been online for just as long, so all other preprints can be regarded as concurrent work. This left the review team with no concerns about novelty or originality. Also, the amount of concurrent preprints clearly indicates that there is sufficient interest in the community about this type of work. Thus, the paper is clearly worthy of acceptance at TMLR.

**Audience:**

The work will be interested for researchers working in the areas of reinforcement learning theory, online learning, and bandit algorithms.

**Claims And Evidence:**

The claims are supported by detailed proofs that were carefully checked and approved by the reviewers.